# Insight into bacterial and archaeal community structure of *Suaeda altissima* and *Suaeda dendroides* rhizosphere in response to different salinity level

Qiqi Wang,[1] Dalun He,[1] Xinrui Zhang,[1] Yongxiang Cheng,[1] Yanfei Sun,[1] Jianbo Zhu[1]

**ABSTRACT**   *Suaeda* as a halophyte with wide adaptability of salinization level plays an important role in saline soil improvement and management. To some extent, the strong salt tolerance of *Suaeda* is influenced by rhizobacteria. However, the effect of different *Suaeda* species and salinization level on the microbial community diversity still unknown. In our study, high-throughput sequencing technology was used to explore the difference in the bacterial and archaeal community diversity of *Suaeda altissima* and *Suaeda dendroides* rhizosphere under high (EC, 8–16 dS/m) and severe (EC > 16 dS/m) saline soil. The result showed that complex soil environment and *Suaeda* species co-shaped bacterial and archaeal community structure, and pH was one of the most important driving factors. In addition, the increase of soil salinity significantly decreased the complexity of bacterial and archaeal co-occurrence network. We explored *Halomonas* and *Candidatus Nitrocosmicus* as core microorganisms in the *Suaeda* rhizosphere, which may play a key ecological role in improving salt tolerance and promoting growth of *Suaeda*. Our study improves the macroscopic understanding of the interrelationships between soil environments-microbes-plants.

**IMPORTANCE**   *Suaeda* play an important ecological role in reclamation and improvement of agricultural saline soil due to strong salt tolerance. At present, research on *Suaeda* salt tolerance mainly focuses on the physiological and molecular regulation. However, the important role played by microbial communities in the high-salinity tolerance of *Suaeda* is poorly studied. Our findings have important implications for understanding the distribution patterns and the driving mechanisms of different Suaeda species and soil salinity levels. In addition, we explored the key microorganisms that played an important ecological role in *Suaeda* rhizosphere. We provide a basis for biological improvement and ecological restoration of salinity-affected areas.

**KEYWORDS**   *Suaeda altissima*, *Suaeda dendroides*, salinization, co-occurrence network, rhizosphere

S oil salinization is buildup of water-soluble salts in soil due to natural formation or human activities, resulting in the degradation or loss of soil productivity function (1). This is a global issue affecting over 100 countries, with approximately 20% of cultivated land and 33% of irrigated land affected by salinization (2, 3). Xinjiang, located in the arid and desert climate of western China, has the largest distribution of saline soil in the country, accounting for over one-fifth of its land area. This makes it a critical region for addressing soil salinization.

Soil salinity inhibits plant growth and leads developmental changes, metabolic adaptations, and ion sequestration or exclusion that rendering a significant portion of agricultural land unusable and reducing productivity, sustainability, and food security

Address correspondence to Yanfei Sun, 81711308@qq.com, or Jianbo Zhu, 274831213@qq.com.

The authors declare no conflict of interest.

See the funding table on p. 17.

(4–7). Abrol (8) classified soil salinization into five categories based on its toxic effects on plants (Table S1). Therefore, effectively restoring salinized soil to maintain ecological security has become a crucial task for protecting ecosystems (9). Halophytes, unlike glycophytes, have stronger salinity adaptability due to their specific oxidation state and ability to resist ion osmotic pressure under salt stress (10). They can also significantly reduce soil salinity and improve soil structure, making them suitable for a wide range of application (11, 12). *Suaeda*, a succulent halophyte of the Chenopodiaceae, can survive and even grow healthily in high-salinity environments (salt concentrations of 200 mM or greater) (13). As a salt-accumulating plant, *Suaeda* facilitates the rapid upward transport of salt to its fleshy leaves, thereby reducing the accumulation of salt in its roots (14). This unique phenomenon of salt accumulation in *Suaeda* is termed "biodrainage" by Heuperman and has been proposed as an effective measure for soil desalination (15, 16). The strong salt tolerance of *Suaeda* is largely related to its complex reaction regulation mechanism. *Suaeda*'s enhanced their salt tolerance through ion regulation and compartmentation, osmotic adjustment of organic solutes, antioxidant capacity regulation, secretion of plant hormones, and changes in the pathway of photosynthetic system (17). Therefore, *Suaeda*'s ability to grow in saline environments and its unique mechanism of salt accumulation make it a crucial player in soil restoration and desalination efforts.

So far, a large amount of basic research has focused on the genes and physiological characteristics related to salt stress in *Suaeda*. Such as, *Suaeda* species form an enhanced transmembrane ion gradient through tonoplast $Na^+/H^+$ antiporter (NHX), vacuolar membrane ATPase ($V-H^+$-ATPase), vacuolar membrane proton pyrophosphatase ($V-H^+$-PPase), $K^+$ transporter, and chloride channels, maintaining the stability of $Na^+$, $K^+$, and $Cl^-$ concentration, to protect *Suaeda* from salt ions (18). However, these researches ignored the rhizosphere microbial contributions to salt stress tolerance of halophytes. A recent eco-physiological approach suggests that the plant-associated microbial community may be the key factor for plants adapted to unfavorable environment (19). *S. salsa* confronted soil salinity by root-microbial interaction, that is an ecological patterned strategy in *S. salsa* system (17).

Rhizosphere is the key niche for interactions among plants, soil, and microorganisms and a complex ecosystem inhabited by numerous microbes including some plant growth-promoting rhizobacteria (PGPR) that participated in material circulation and enhanced plant tolerance to biotic and abiotic stress (20). Studies have shown that PGPR (such as *Pantoea agglomerans*, *Bacillus* sp., and *Sphingobacterium*) can relieve salt stress by increasing content of proline, slowing down senescence, maintaining the ion balance, and reducing reactive oxygen species in plants (such as *Casuarina obesa* and tomato) (21–23). Furthermore, archaea, as an important microbial population, can live in environments with extreme conditions, and they play an important ecological role in the plant rhizosphere. Ammonia-oxidizing archaea (AOA) mediates soil N-cycle to support plant growth and health (24). Many members of archaea contain alkaline phosphatases PhoD and PhoX, which hydrolyze soil organic-P, that increased available phosphorus content in the rhizosphere of plants (25). Besides environmental nutrient cycling and promoting plant growth in plant ecosystems, archaea also enhance abiotic stress resistance of plants (26). The metagenome analysis of archaea from alpine bogs suggested functional potential in protecting plants from oxidative and osmotic stresses (27). Thus, the halophyte microbiome plays a key function in its high tolerance to excessive soil salinity for a better wild crop production. We speculate that there are also a large number of bacteria and archaea in the rhizosphere of *Suaeda*, which play an important role in alleviating salt stress and promoting plant growth. Therefore, exploring beneficial microbial resources in the rhizosphere of *Suaeda* is crucial for the development of agricultural microbe-agent to improve plant stress resistance.

To clarify the interaction mechanisms and ecological functions of plant-associated microbiota, the diversity, and influencing factors of rhizosphere microorganisms, it is necessary to be investigated. Previous studies indicated that soil pH value, soil

temperature, plant species, soil cultivation, and salinity stress can directly or indirectly affect the activities of rhizosphere microorganisms (28). Soil salinity, as one of the important influencing factors, has been highly concerned in recent years. Soil high salinity significantly decreased the abundance of nitrite-oxidizing bacteria and inhibited the nitrite oxidation rates (29). Salinity also was the driving force on the distribution and diversity of anammox consortium, improved the quorum sensing in anammox consortium, and increased the bacterial tolerance to salt stress (30). It can be seen that soil salinity affects the function of soil microbial communities, which adverse to biogeochemical nitrogen cycling. Gao assessed the effect of salt stress on sorghum growth performance and rhizosphere bacterial community structure, and their results showed that soil bacterial community responses to salinity and plant root exudation could potentially impact the microenvironment to help plants overcome external stressors and promote sorghum growth (31). On the other hand, plants adapt to biotic stress by changing their root exudation chemistry to assemble health-promoting microbiomes (32). This was be called "cry-for-help" hypothesis, and plants also can reprogram the functional expression of inhabited rhizobacteria to improve their adaptation and resistance to the stress (33, 34). However, the composition of root exudates can vary among plant species, leading to the rhizosphere microbial community that varies due to different microbial preferences for root exudates (35). We hypothesized that there are also differences in the structure of the rhizosphere microbial communities of different *Suaeda* species under salt stress. However, differences in the response mechanisms of *S. altissima* and *S. dendroides* in response to salt-stressed environments have not been reported.

In our study, we hypothesis that (i) the soil salinity level and *Suaeda* species jointly shape the structure of the rhizosphere bacterial and archaeal community and (ii) the core microbial communities in the *Suaeda* rhizosphere that play an important ecological role in alleviating salt tolerance of *Suaeda*. In this study, our objectives are (i) investigated the impact of high and severe salinity levels on the bacteria and archaea diversity and community structure, (ii) investigated the influence of *S. altissima* and *S. dendroides* on bacteria and archaea diversity and community structure, and (iii) explored the core bacteria and archaea associated with enhancing the salt tolerance of *Suaeda*. Our findings provide insights into the connection between rhizosphere microorganisms and salt tolerance of *Suaeda*, which could have practical implications for improving plant growth in saline soils.

## RESULTS

### Environmental characteristics of rhizosphere samples from *S. altissima* and *S. dendroides*

The results of environment physicochemical parameters have been showed in Table 1. GJP-6 soil samples exhibited significantly higher levels of TN, TP, TK, AHN, AP, AK, and $K^+$ compared to soil samples from other three locations. The pH value of all of soil samples exceeded 7.0, with MJP-2 recording the highest pH value of 8.95. EC values exhibited significant difference across soil samples, with MJP-6 displaying the highest value, followed by MJP-5, MJP-2, and MJP-3. In addition, MJP-5 had the highest content of $Na^+$, $Cl^-$, $SO_4^{2-}$, $Ca^{2+}$, and $Mg^{2+}$. In contrast, MJP-2 had the lowest values of OM, TP, TK, AP, and $HCO_3^-$. Out of the 16 geochemical properties measured, $Cl^-$, $SO_4^{2-}$, $Ca^{2+}$, and $Na^+$ content differed significantly between the rhizosphere soil of *S. dendroides* (MJP-2 and MJP-5) and *S. altissima* (GJP-3 and GJP-6). On the other hand, soil samples from the rhizosphere of *S. altissima* exhibited significantly higher values of OM, TN, TP, AHN, and AP content ($P < 0.05$) than those from rhizosphere of *S. dendroides* (Table 1). MJP-5 displayed the highest MBC value (0.33 mg/g), activity of sucrase (73.95 mg/g), and activity of phosphate reductase (0.149 mg/g). In addition, soil samples from GJP-3 have the strongest activity of phosphatase (0.88 mg/g), while the highest activity of soil urease was observed in GJP-6 samples.

**TABLE 1** Environmental characteristics of rhizosphere soil samples from *Suaeda dendroides* and *Suaeda altissima* (expressed as mean value and standard error)[a]

|  | MJP-2 | MJP-5 | GJP-3 | GJP-6 |
|---|---|---|---|---|
| OM (g/kg) | 9.28 ± 0.40c | 9.18 ± 0.19c | 30.3 ± 0.49a | 16.82 ± 0.31b |
| TN (g/kg) | 0.56 ± 0.01b | 0.50 ± 0.02c | 1.08 ± 0.02a | 1.05 ± 0.03a |
| TP (g/kg) | 0.74 ± 0.01c | 0.76 ± 0.01bc | 0.78 ± 0.02b | 1.09 ± 0.00a |
| TK (g/kg) | 21.00 ± 0.23c | 21.93 ± 0.20b | 15.94 ± 0.16d | 29.55 ± 0.51a |
| AHN (mg/kg) | 24.96 ± 0.36d | 38.83 ± 1.44c | 215.94 ± 3.05b | 262.60 ± 1.35a |
| AP (mg/kg) | 4.60 ± 0.19c | 4.14 ± 0.21c | 14.22 ± 0.1b | 95.68 ± 0.50a |
| AK (mg/kg) | 325.23 ± 6.65bc | 347.20 ± 10.22b | 302.03 ± 1.33c | 1,273.33 ± 23.97a |
| pH | 8.95 ± 0.01a | 8.87 ± 0.01b | 8.23 ± 0.01d | 8.34 ± 0.01c |
| EC (dS/m) | 14.19 ± 0.12b | 17.05 ± 0.56a | 13.15 ± 0.38b | 16.57 ± 0.61a |
| $Cl^-$ (mg/g) | 3.32 ± 0.10a | 3.48 ± 0.14a | 1.47 ± 0.21b | 0.90 ± 0.03c |
| $SO4^{2-}$ (mg/g) | 8.96 ± 0.63b | 13.12 ± 0.16a | 5.47 ± 0.58c | 5.39 ± 0.16c |
| $Ca^{2+}$ (mg/g) | 2.79 ± 0.13a | 2.70 ± 0.07a | 1.48 ± 0.16c | 1.15 ± 0.05d |
| $K^+$ (mg/g) | 0.14 ± 0.00b | 0.03 ± 0.00c | 0.08 ± 0.01d | 0.56 ± 0.01a |
| $Mg^{2+}$ (mg/g) | 0.28 ± 0.00b | 0.36 ± 0.01a | 0.33 ± 0.04ab | 0.15 ± 0.01c |
| $Na^+$ (mg/g) | 3.89 ± 0.20b | 7.24 ± 0.27a | 2.22 ± 0.11c | 2.28 ± 0.08c |
| $HCO_3^-$ (mg/g) | 0.11 ± 0.00c | 0.16 ± 0.01b | 0.18 ± 0.01a | 0.12 ± 0.01c |
| MBC (mg/g) | 0.17 | 0.33 | 0.28 | 0.26 |
| CAT (mg/g) | 5.48 | 5.65 | 6.08 | 4.86 |
| URA (mg/g) | 0.40 | 0.19 | 0.51 | 0.71 |
| PRO (mg/g) | 0.24 | 0.11 | 0.12 | 0.31 |
| PHO (mg/g) | 0.39 | 0.13 | 0.88 | 0.09 |
| SUC (mg/g) | 40.16 | 73.95 | 48.97 | 70.25 |
| NIT (mg/g) | 0.065 | 0.149 | 0.016 | 0.026 |

[a]The different lowercase letters indicate the significant difference ($P < 0.05$) in the same environmental factor among MJP-2, MJP-5, GJP-3, and GJP-6 based on one-way ANOVA followed by Duncan test. OM, organic matter; TN, total nitrogen; TP, total phosphorus; TK, total potassium; AHN, available nitrogen; AP, available phosphorus; AK, available potassium; EC, electrical conductivity, MBC, microbial biomass carbon; CAT, catalase; URE, urease; PRO, protease; SUC, Sucrase; PHO, Phosphate reductase; NIT, nitrite reductase.

## Sequencing data characteristics of rhizosphere samples from *S. altissima* and *S. dendroides*

We analyzed 12 rhizosphere soil samples and obtained 587,427 high-quality sequences, with reads ranging from 30,845 to 71,840 and an average length of 276 bp. After rarefaction to 48,951.5 sequences per sample (Table 2), we clustered the sequences into 3,877 OTUs (bacteria, 3,722 OTUs; archaea, 155 OTUs). These OTUs were classified into 38 phyla; 121 classes; 286 orders; 499 families; 889 genera; and 1,468 bacterial species and 7 phyla, 10 classes, 13 orders, 21 families, 46 genera, 71 archaeal species, respectively. We found 1,436 common bacterial and 19 common archaeal OTUs across all samples, as shown in the Venn diagram. The numbers of unique OTUs in MJP-2, MJP-5, GJP-3, and GJP-6 samples were 237, 284, 53, and 97 for bacteria and 30, 11, 7, and 0 for archaea (Fig. 1). In all soil samples, bacterial OTUs outnumbered archaeal OTUs. Rarefaction curves (Fig. S1) indicated that sequencing libraries sufficiently covered the diversity of most bacterial and archaeal species in samples analyzed for this study, ensuring the validity and reliability of our results. However, during data processing, we observed a significant difference in the result of GJP3-1 compared to the other samples (Fig. S2). As a result, we excluded this sample from subsequent calculations and analyses.

## Bacterial and archaeal community composition of rhizosphere samples from *S. altissima* and *S. dendroides*

After removing outliers, we observed higher ACE and Chao1 indices in both bacterial and archaeal communities in samples from MJP-2 and GJP-3. Among bacteria, the Shannon index was higher in MJP-5 and GJP-6 compared to MJP-2 and GJP-3. For archaea, the

**TABLE 2** The statistic results of sequence information from rhizosphere soil samples of *Suaeda altissima* and *Suaeda dendroides*[a]

| Sample information | Sequence number | Base number | Mean length | Min length | Max length |
|---|---|---|---|---|---|
| MJP2-1 | 71,640 | 19,781,781 | 276.1275963 | 203 | 320 |
| MJP2-2 | 57,022 | 15,753,279 | 276.2666865 | 205 | 314 |
| MJP2-3 | 41,518 | 11,467,933 | 276.2159304 | 200 | 321 |
| **MJP-2** | **56,726.67 ± 8,207.03a** | **15,667,664.33 ± 4,157,585a** | **276.20 ± 0.07a** | **202.67 ± 2.52a** | **318.33 ± 3.79b** |
| MJP5-1 | 70,005 | 19,328,144 | 276.0966217 | 205 | 349 |
| MJP5-2 | 31,171 | 8,608,107 | 276.1575503 | 201 | 355 |
| MJP5-3 | 36,863 | 10,180,693 | 276.1764642 | 209 | 311 |
| **MJP-5** | **46,013.00 ± 20,971.69a** | **12,705,648 ± 5,788,899a** | **276.14 ± 0.04ab** | **205.00 ± 4.00a** | **338.33 ± 23.86ab** |
| GJP3-1 | 55,915 | 15,432,438 | 275.9981758 | 212 | 357 |
| GJP3-2 | 30,854 | 8,517,005 | 276.1227103 | 209 | 345 |
| GJP3-3 | 66,989 | 18,489,521 | 276.0083148b | 201 | 346 |
| **GJP-3** | **51,249.67 ± 18,518.13a** | **1,414,632.00 ± 5,109,143.00a** | **276.04 ± 0.07b** | **207.33 ± 5.69a** | **349.33 ± 6.66a** |
| GJP6-1 | 41,790 | 11,541,375 | 276.1755205 | 229 | 327 |
| GJP6-2 | 33,623 | 9,284,122 | 276.1241412 | 202 | 343 |
| GJP6-3 | 50,037 | 13,817,788 | 276.151408ab | 202 | 362 |
| **GJP-6** | **41,816.67 ± 8,207.03a** | **11,547,762.00 ± 2,266,840.00a** | **276.20 ± 0.07a** | **202.67 ± 2.52a** | **318.33 ± 3.79ab** |

[a]Black font values represent mean ± standard deviation. The different letters indicate the significant difference ($P < 0.05$) in the same item among MJP-2, MJP-5, GJP-3, and GJP-6 based on one-way ANOVA followed by Duncan test.

Shannon index was significantly higher in MJP-2 and MJP-5 compared to GJP-3 and GJP-6, while the Chao1 and ACE indices were higher in MJP-2 and GJP-3 compared to MJP-5 and GJP-6. Additionally, we noted that the alpha diversity indices of the bacterial community were significantly higher than archaeal community (Table 3).

At the phylum level, barplot analysis demonstrated the diversity of bacterial communities across different samples. We identified 13 bacterial and 5 archaeal phyla, each with a relative abundance greater than 1%. Phyla with a relative abundance less than 0.01% were merged into "other" group (Fig. 2). For bacteria, Actinobacteriota and Proteobacteria were the dominant phyla, with a relative abundance greater than 20%. They accounted for 32.72% (GJP-6) to 48.85% (MJP-5) and 24.47% (GJP-3) to 32.64% (MJP-5) across all soil samples. Bacteroidota, Chloroflexi, and Gemmatimonadota followed closely behind (Fig. 2A). For archaea, Crenarchaeota, Thermoplasmatota, and Halobacterota were the dominant phyla, accounting for an average of 49.48% (MJP-2) to 92.80% (GJP-6), 1.38% (GJP-3) to 29.22% (MJP-2), and 0.45% (GJP-6) to 20.70% (MJP-2) of all samples. We observed that Thermoplasmatota and Halobacterotan were the more abundant in MJP-2 and MJP-5 than in GJP-3 and GJP-6. However, Crenarchaeota accounted for a higher abundance in GJP-3 and GJP-6 (Fig. 2B).

The heatmap analysis revealed the top 25 bacterial and archaeal genera by relative abundance (Fig. 3). Among the bacterial genera, *norank_f_norank_o_Acidobacteriales* (6.51)%, *norank_f_norank_o_norank_c_Alphaproteobacteria* (3.68%), *Halomonas* (3.12%), *norank_f_67–14* (2.38%), *norank_f_Geminicoccaceae* (2.12%), and *Streptomyces* (2.07%) had a relative aboundance more than 2% in all samples (Fig. 3A). *norank_f_Balneolaceae* and *norank_f_Nitriliruptoraceae* were more abundant in MJP-2 and MJP-5, whereas *norank_f_67–14* and *Streptomyces* showed heigher relative abundance in GJP-3 and GJP-6. In addition, *norank_f_Euzebyaceae* and *norank_f_ norank_o_Chloroplast* were more abundant in MJP-2 and GJP-3, while *Halomonas* was more richness in MJP-5 and GJP-6. These dominant bacterial genera all belonged to Actinobacteriota. For archaeal, *Candidatus Nitrocosmicus* (54.61%), *unclassified_c_Thermoplasmata* (12.12%), and *norank_f_Nitrososphaeraceae* (7.08%) had a relative aboundance more than 2% in all samples (Fig. 3B). *Candidatus_Nitrocosmicus* was the most abundant genus due to its high expression. *Halolamina, unclassified_c_Thermoplasmata, norank_f_Halomicrobiaceae, Saliphagus,* and *Haladaptatus* were significantly more abundant and frequent in MJP-2 and MJP-5 than GJP-3 and GJP-6.

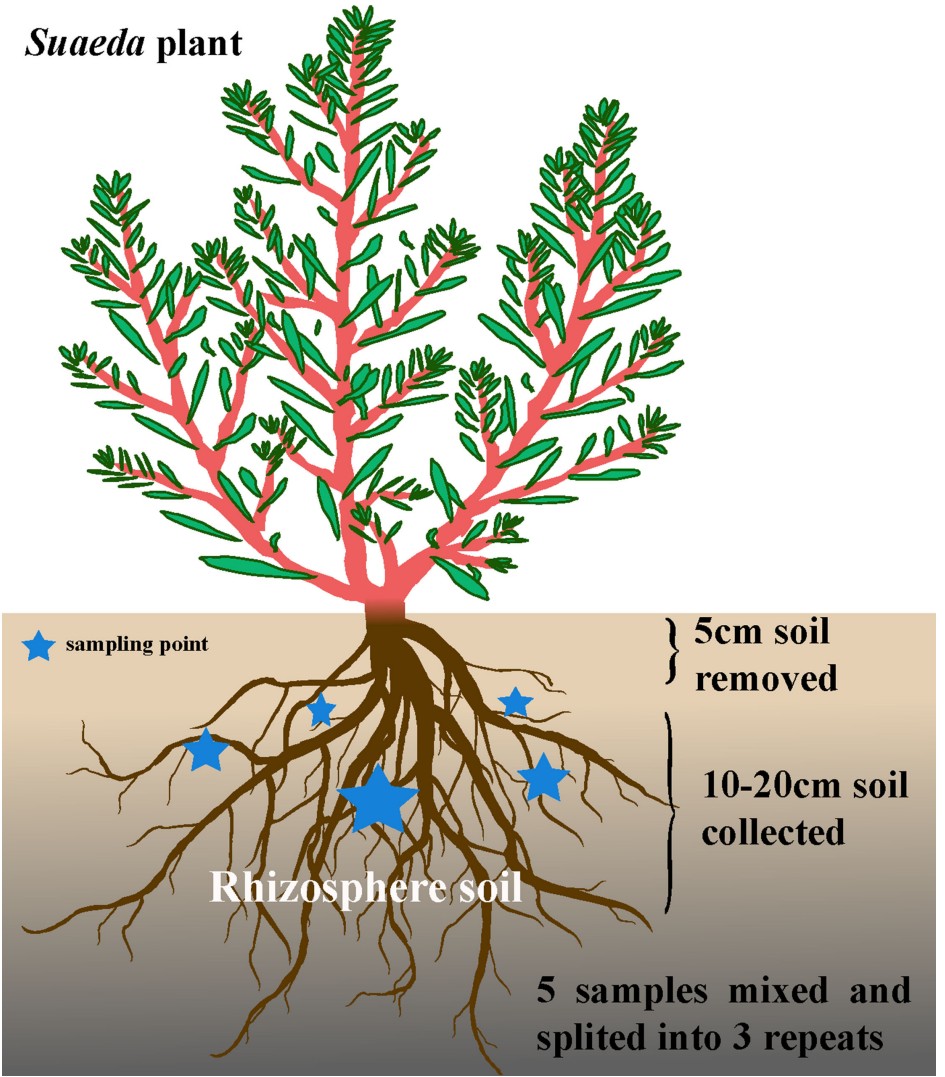

*Suaeda* plant

sampling point

5cm soil removed

10-20cm soil collected

Rhizosphere soil

5 samples mixed and splited into 3 repeats

**FIG 1** Model diagram of collection method of soil samples in *Sueada* rhizosphere.

## Comparative analysis of significantly different bacterial and archaeal genus among different samples

We used the Kruskal-Wallis H test to identify bacterial genera with significant differences in richness across soil samples (Fig. 4). Out of 889 bacterial genera, we found 10 that showed significant differences: *Marmoricola* ($P = 0.01396$), *norank_f_norank_o_norank_c_Alphaproteobacteria* ($P = 0.00433$), *norank_f_norank_o_norank_c_S0134_terrestrial_group* ($P = 0.00067$), *norank_f_Gemmatimonadaceae* ($P = 0.02384$), *Sphingomonas* ($P = 0.04841$), *Glycomyces* ($P = 0.02113$), *Ralstonia* ($P = 0.04841$), *unclassified_f_Nocardioidaceae* ($P = 0.01382$), *norank_f_Rhodothermaceae* ($P = 0.0003358$), and *Pontibacter* ($P = 0.00403$). All these genera showed significantly different across all samples. Specifically, *Marmoricola*, *norank_ f_ rank_ o_ norank_ c_ S0134_ terrestrial_ Group*, *Glycomics* were significantly enriched in MJP-2 and GJP-3 samples (high salinity level), while *norank_ f_ Gemmatimonadaceae*, *Sphingomonas*, and *unclassified_f_Nocardioidaceae* were significantly enriched in GJP-3 and GJP-6 (*S. altissima*) (Fig. 4A). Regarding archaea, we observed significant differences in only 2 of the 46 genera: *unclassfied_ p_ Thermoplastota* and *Halobacterium*. These genera were significantly enriched in MJP-2 and MJP-5 samples (*S. dendroides*) (Fig. 4B).

**TABLE 3** Alpha diversity index of bacterial and archaeal community in rhizosphere soil of *Suaeda dendroides* and *Suaeda altissima*[a]

| | Sample ID | Sobs index | Shannon index | Simpson index | ACE index | Chao1 index | Good's coverage |
|---|---|---|---|---|---|---|---|
| Bacteria | MJP-2 | 1,894 ± 438.41a | 5.86 ± 0.54a | 0.012 ± 0.00a | 2,237.03 ± 456.92a | 2,264.11 ± 465.42a | 0.99 ± 0.00a |
| | MJP-5 | 1,752 ± 691.34a | 6.04 ± 0.26a | 0.008 ± 0.00a | 2,009.73 ± 866.14a | 2,026.68 ± 876.88a | 0.99 ± 0.01a |
| | GJP-3 | 1,733 ± 187.38a | 6.04 ± 0.12a | 0.007 ± 0.00a | 2,133.71 ± 68.64a | 2,175.46 ± 70.24a | 0.99 ± 0.01a |
| | GJP-6 | 1,568 ± 108.28a | 5.95 ± 0.61a | 0.009 ± 0.00a | 1,836.98 ± 28.64a | 1,886.69 ± 53.24a | 0.99 ± 0.00a |
| Archaea | MJP-2 | 99.67 ± 18.23a | 3.10 ± 0.30a | 0.140 ± 0.06b | 111.48 ± 17.36a | 109.16 ± 20.70a | 0.99 ± 0.01a |
| | MJP-5 | 45.67 ± 20.60b | 2.84 ± 0.61a | 0.107 ± 0.07b | 57.39 ± 25.25ab | 52.33 ± 27.00b | 0.99 ± 0.01a |
| | GJP-3 | 47.5 ± 43.13b | 1.53 ± 0.91b | 0.442 ± 0.22a | 67.38 ± 51.10ab | 65.71 ± 47.68ab | 0.99 ± 0.01a |
| | GJP-6 | 19.67 ± 9.81b | 1.57 ± 0.12b | 0.305 ± 0.07ab | 25.11 ± 10.02b | 22.05 ± 12.22b | 0.99 ± 0.00a |

[a]The different letters indicate the significant difference ($P < 0.05$) in the same item among MJP-2, MJP-5, GJP-3, and GJP-6 based on one-way ANOVA followed by Duncan test.

## Comparative analysis of the similarities and difference in community composition among different samples

We conducted hierarchical clustering using Weighted UniFrac distance to compare the bacterial and archaeal communities in the rhizosphere of *S. altissima* and *S. dendroides*. Our result revealed two distinct groups based on bacterial community structure, with MJP-2 and MJP-5 forming one group and samples GJP-3 and GJP-6 forming another (Fig. 5A). Meanwhile, archaeal communities showed a similar pattern (Fig. 5C). These findings suggest that the bacterial and archaeal community compositions of *S. altissima* and *S. dendroides* are significantly different. Adonis analysis confirmed these differences, with siginifcant variation in both bacterial ($R^2 = 0.45$, $P = 0.045*$) and archaeal ($R^2 = 0.41$, $P = 0.098$) community structure among samples MJP-2, MJP-5, GJP-3, and GJP-6 (Table S3).

## The relationship among the domain bacterial and archaeal genus, environmental characteristics, and soil samples of *Suaeda* rhizosphere

Our analysis using Pearson correlation revealed a moderate correlation between soil characteristics and alpha diversity index of bacteria though this was not statistically significant. However, for archaea, we found a significant negative correlation between Shannon, Simpson, ACE, and Chao1 indices and soil OM, TN, TP, TK, AHN, AP, K$^+$, and URE (ranging from −0.940 to −0.392). Conversely, we observed a significant positive correlation between ACE and Chao1 indices and but pH, Cl$^-$, Mg$^{2+}$, Na$^+$, SO$_4^{2-}$, and

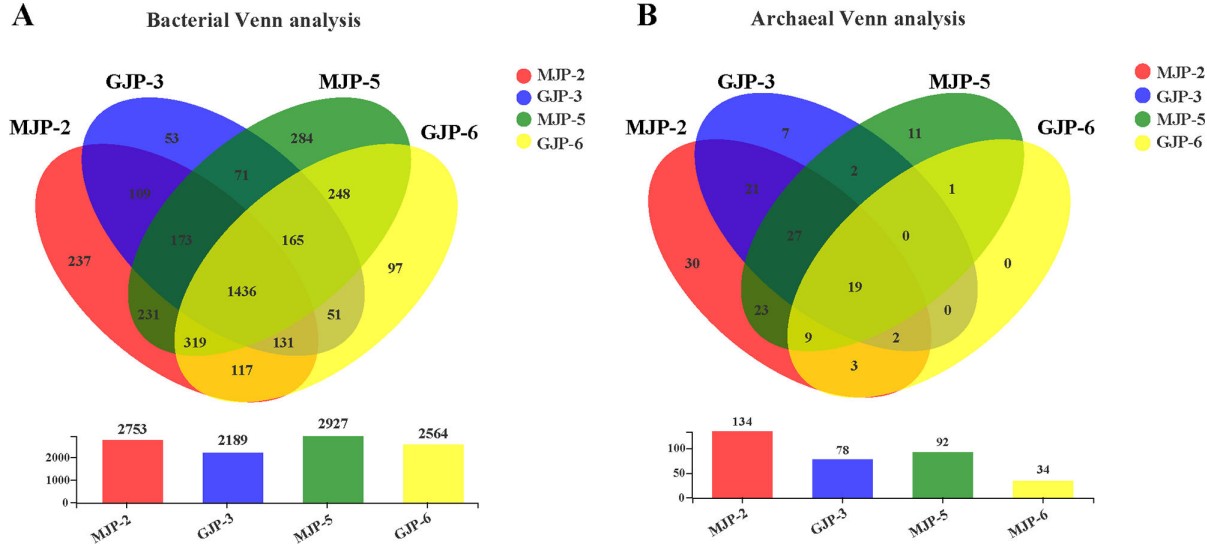

**FIG 2** Venn analysis of share and unique OTU of bacterial and archaeal in different soil samples of *Suaeda dendroides* and *Suaeda altissima*. Bar chart indicating the total number of OTUs contained in each sample. A, bacteria; B, archaea.

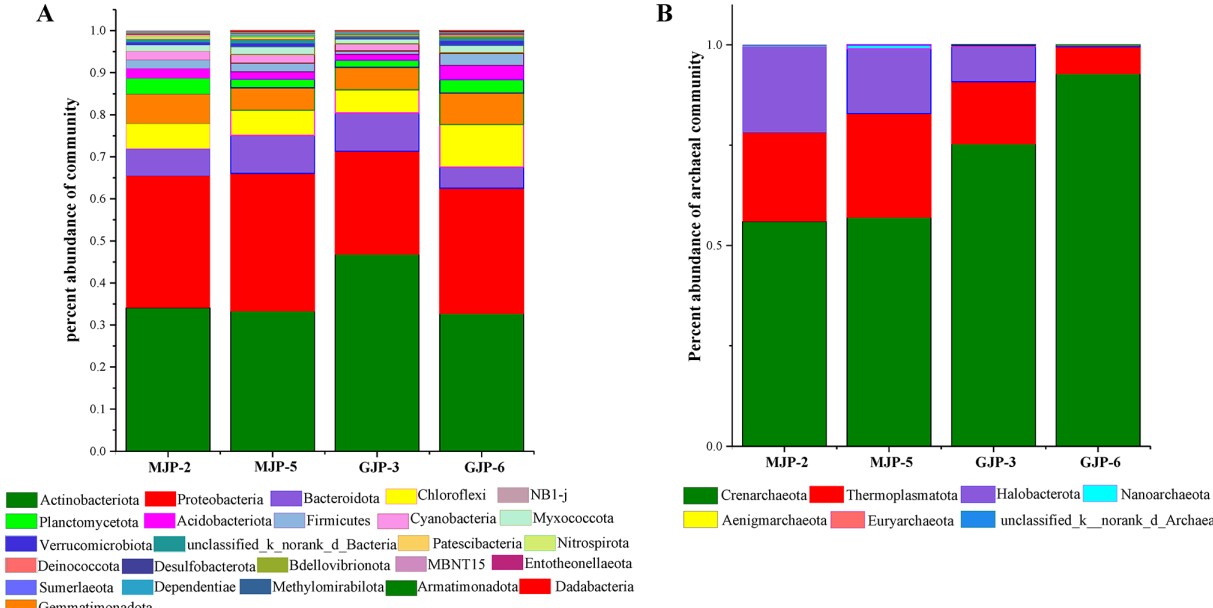

**FIG 3** Barplot analysis of bacterial (A) and archaeal (B) community abundance of rhizosphere soil of *Suaeda dendroides* and *Suaeda altissima*. in phylum level.

$Ca^{2+}$. Moreover, CAT was significantly negatively correlated with Shannon index (ranging from 0.578 to 0.929) (Table S4). The results indicate a strong association between soil properties and the microbial community, particularly for archaea. The findings suggest that certain soil properties may influence the diversity and composition of archaea in soil.

To explore the relationship among domain 10 genus, environment factors, and soil samples from rhizosphere of *S. altissima* and *S. dendroides*, redundancy analysis (RDA) (Fig. 5B and D) and Monte Carlo permutation tests were performed. The bacterial community distribution was found to be significantly associated with pH ($R^2 = 0.499$, $P = 0.046^*$) and SUC ($R^2 = 0.571$, $P = 0.035^*$) (Table 4), as shown by RDA1 and RDA2, which explained 52.36% and 12.80% of the total variation, respectively (Fig. 5B). Samples of MJP-2 and MJP-5 were found to be closely correlated with SUC, EC, $Na^+$, and pH, while samples of GJP-3 and GJP-6 were positively correlated

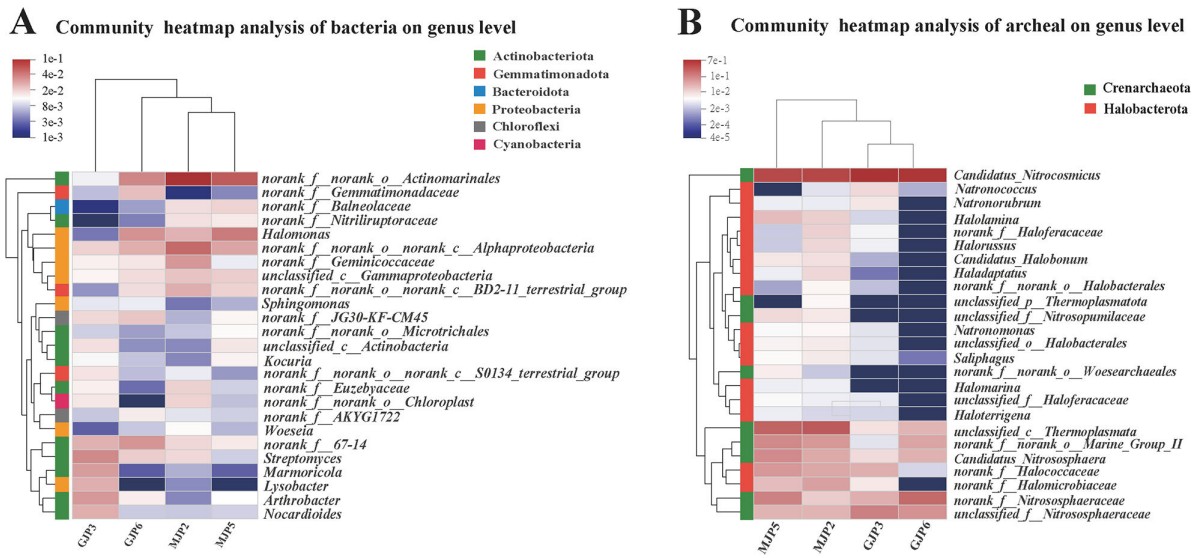

**FIG 4** Heatmap analysis of bacterial (A) and archaeal (B) community structure in genus level.

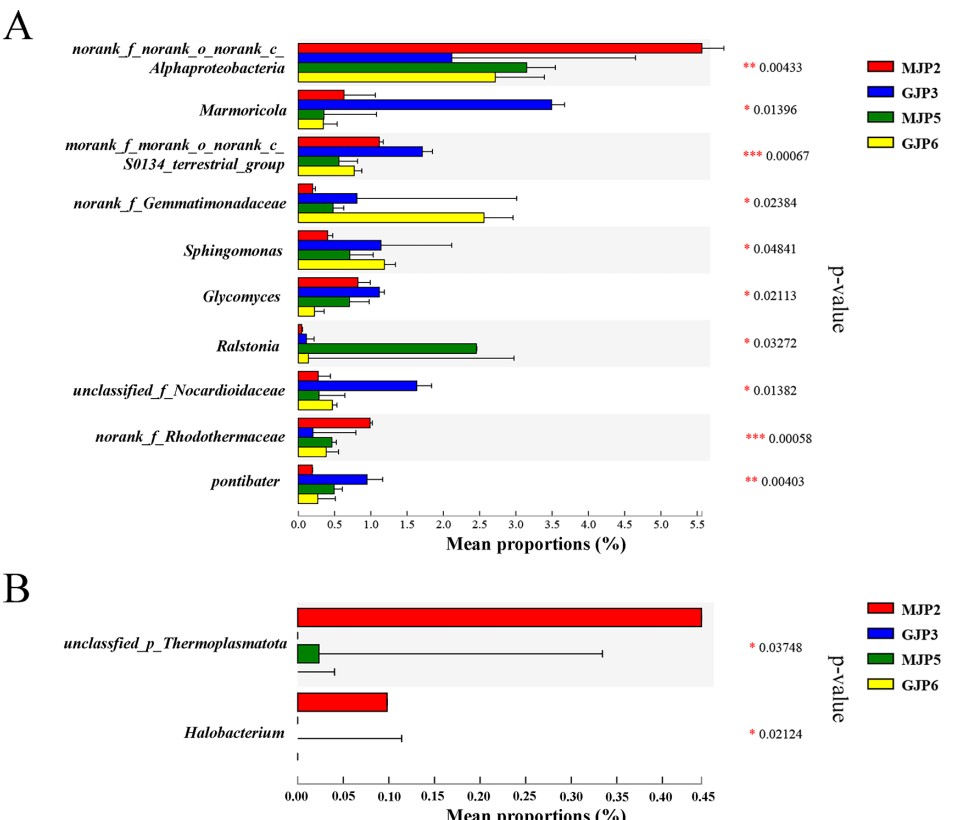

**FIG 5** Species difference analysis of the rhizosphere soil samples of *Suaeda dendroides* and *Suaeda altissima* in genus level: (A) bacteria, (B) archaea. The y-axis represents the classification levels of species, and the x-axis represents the percentage of species average relative abundance in each sample group. The Kruskal-Wallis rank-sum test was used to show significant differences (*: $0.01 < P <= 0.05$, **: $0.001 < P <= 0.01$, ***: $P <= 0.001$).

with SUC, PHO, URE, $HCO_3^-$, and $K^+$ but negatively correlated with EC, PH, and $Na^+$. Moreover, *norank_f_JG30-KF-CM45*, *Arthrobacter*, *Streptomyces*, and *norank_f_67–14* were found to be closed correlated with PHO, URE, and $HCO_3^-$, while *norank_f_Gemini-coccaceae*, *norank_f_norank_o_norank_c_Alphaproteobacteria*, *unclassified_c_Gammap-roteobacteria*, *norank_f_norank_o_norank_c_BD2-11_terrestrial group*, *Halomonas,* and *norank_f_norank_o_Actinomarina* were found to be significantly correlated with $Na^+$, EC, and pH (Fig. 5B). In summary, the results of our study suggest that pH and SUC have a significant impact on the bacterial community distribution in the rhizosphere of *S. altissima* and *S. dendroides*.

In terms of archaea, two factors, RDA1 and RDA2, accounted for 81.39% and 15.52% of the total variation, respectively (Fig. 5D). Our analysis revealed a significant relationship between the distribution of archaeal communities and pH ($R^2 = 0.628$, $P = 0.022*$) and PHO ($R^2 = 0.539$, $P = 0.049*$) (Table 4). Notably, samples of MJP-2 and MJP-5 were positively associated with $Na^+$, EC, and pH but negatively associated URE, PHO, and $K^+$. In contrast, samples of GJP-3 and GJP-6 showed a positively correlation with PHO, URE, and $K^+$. Additionally, *unclassified_c_Thermpplasmata* appeared to be closely associated with pH, and *Candidatus Nitrocosmicus* exhibited a strong correlation with URE, PHO, and $K^+$ (Fig. 5D).

## Effects of soil salinity and *Suaeda* species on the co-occurrence network of the rhizosphere bacterial and archaeal communities

To investigate the impact of soil salinity and *Suaeda* species on soil bacterial and archaeal communities, we analyzed the correlation between OTUs with high abundance (top 50)

**TABLE 4** The correlation among environmental factors and microbial community[a]

| | Bacteria | | | | Archaea | | | |
|---|---|---|---|---|---|---|---|---|
| | RDA1 | RDA2 | $R^2$ | $P$ values | RDA1 | RDA2 | $R^2$ | $P$ values |
| TK | 0.126 | 0.992 | 0.295 | 0.251 | 0.246 | 0.969 | 0.044 | 0.798 |
| pH | 0.993 | 0.119 | 0.499 | 0.046[b] | 0.511 | 0.860 | 0.628 | 0.022[b] |
| EC | 0.708 | 0.707 | 0.263 | 0.289 | 0.861 | 0.510 | 0.429 | 0.098 |
| K[+] | 0.700 | 0.715 | 0.130 | 0.605 | 0.658 | 0.753 | 0.238 | 0.379 |
| Na[+] | 0.671 | 0.742 | 0.215 | 0.375 | 0.885 | 0.465 | 0.412 | 0.112 |
| HCO₃[−] | 0.957 | 0.292 | 0.199 | 0.426 | 0.214 | 0.977 | 0.168 | 0.489 |
| URE | 0.989 | 0.145 | 0.144 | 0.541 | 0.757 | 0.654 | 0.411 | 0.112 |
| PHO | 0.970 | 0.242 | 0.309 | 0.225 | 0.671 | 0.742 | 0.539 | 0.049 |
| SUC | 0.431 | 0.902 | 0.571 | 0.035[b] | 0.536 | 0.844 | 0.254 | 0.267 |

[a]Structure of rhizosphere soil from *Suaeda altissima* and *Suaeda dendroides* by Monte Carlo permutation tests.
[b]Indicates a significant correlation, $P < 0.05$, ** indicates an extremely significant correlation, $P < 0.01$. TK, total potassium; EC, electrical conductivity; URE, urease; SUC, sucrase; PHO, phosphate reductase.

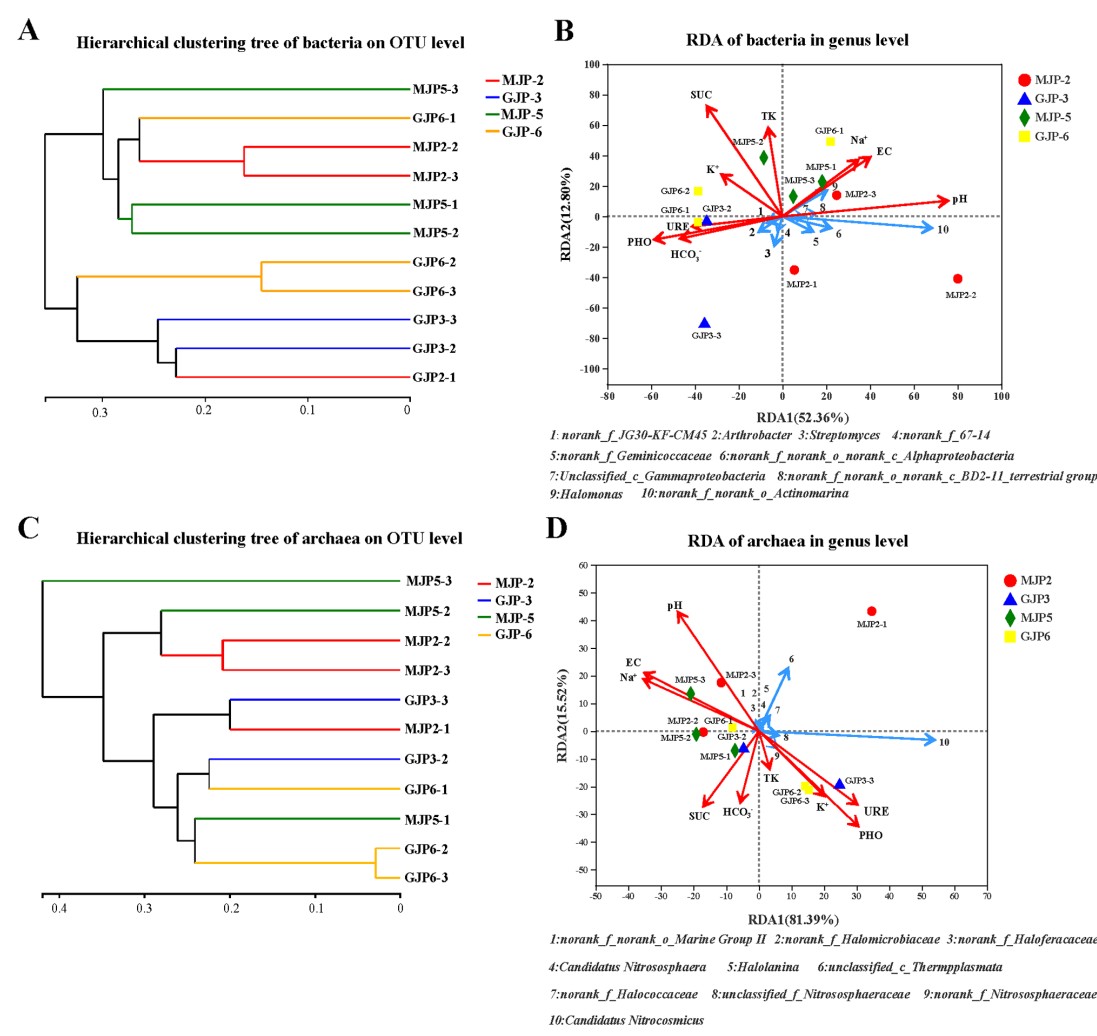

**FIG 6** Hierarchical clustering tree were used to analysis the similarity of bacterial (A) and archaeal (C) community structure among samples from rhizosphere soil samples of *Suaeda dendroides* and *Suaeda altissima*. Redundancy analysis (RDA) ordination biplot were used to explore the relationships among bacteria (B) and archaea (D) abundance, soil properties and the top 10 genu of bacteria and archaea of the rhizosphere soil samples from *Suaeda dendroides* and *Suaeda altissima*. pH, soil acidity; EC, electric conductivity; Na[+], soil sodium ion content; K[+], soil potassium ion content; HCO₃[−], soil bicarbonate content; TK, soil total potassium; PHO, phosphatase; URE, urease; SUC, sucrase; The Monte Carlo permutation test of the RDA was performed.

using Spearman correlation analysis. OTUs with a significant correlation ($|r| > 0.6$ and $P < 0.05$) were selected for further analysis. We also constructed a co-occurrence network to visualize the composition of bacterial and archaeal communities under different soil salinity and *Suaeda* species (Fig. 6).

For bacteria, the number of edges (that is, degree) of the microbial network in MJP-2, MJP-5, GJP-3, and GJP-6 varied among bacterial species, with values of 872; 768; 1,764; and 1,346, respectively. The edges number of MJP-2, MJP-5, GJP-3, and GJP-6 were 48, 49, 49, and 48. In addition, we observed a decrease in the complexity of rhizosphere microbial community of the same *Suaeda* species with increasing of soil salinity. The number of edges the network of *S. altissima* and *S. dendroides* decreased 23.70% and 11.93%, respectively (Fig. 6A; Table S5A through D). The identifiable dominant OTUs were primarily assigned to Actinobacteriota, Proteobacteria, Gemmatimonadota, and Cyanobacteria. We found that OTU420 (*norank_f__norank_o__norank_c__Alphaproteobacteria*), OTU1792 (*norank_f__norank_o__Actinomarinales*), OTU374 (*Halomonas*), OTU839 (*g__norank_f__norank_o__Actinomarinales*), OTU35 (*Arthrobacter*), and OTU1757 (*Marmoricola*) exhibited higher value of degree centrality, closeness centrality, and betweenness centrality (Fig. 7A; Table S6A).

For archaea, the edge numbers of network in MJP-2, MJP-5, GJP-3, and GJP-6 were 1,022; 522; 1,184; and 390; respectively. The edge numbers of MJP-2, MJP-5, GJP-3, and GJP-6 were 48, 50, 30, and 31 (Fig. 6B; Table S5E through H). Surprisingly, we observed that soil salinity had a greater impact on the complexity of archaeal communities than on bacterial communities. The identifiable dominant OTUs were primarily assigned to Halobacterota, Thermoplasmatota, and Crenarchaeota. In addition, OTU91 (*Candidatus_Nitrocosmicus*), OTU3255 (*Candidatus_Nitrocosmicus*), OTU239 (*unclassified_f_Nitrososphaeraceae*), OTU2405 (*norank_f_Halococcaceae*) exhibiting higher values of degree centrality, closeness centrality, and betweenness centrality in all samples (Fig. 7B; Table S6B). Furthermore, we found that soil salinity increased the negative correlation between soil microorganisms, including both bacteria and archaea.

## DISCUSSION

The halophytic plant-associated microbial community and halotolerant PGPRs play important roles in allowing hosts to adapt to a costal environment (36). They play a crucial role in nutrient cycling, organic matter decomposition, and enhancing plant resistance to abiotic and biotic stress to maintain productivity (37). However, the diversity

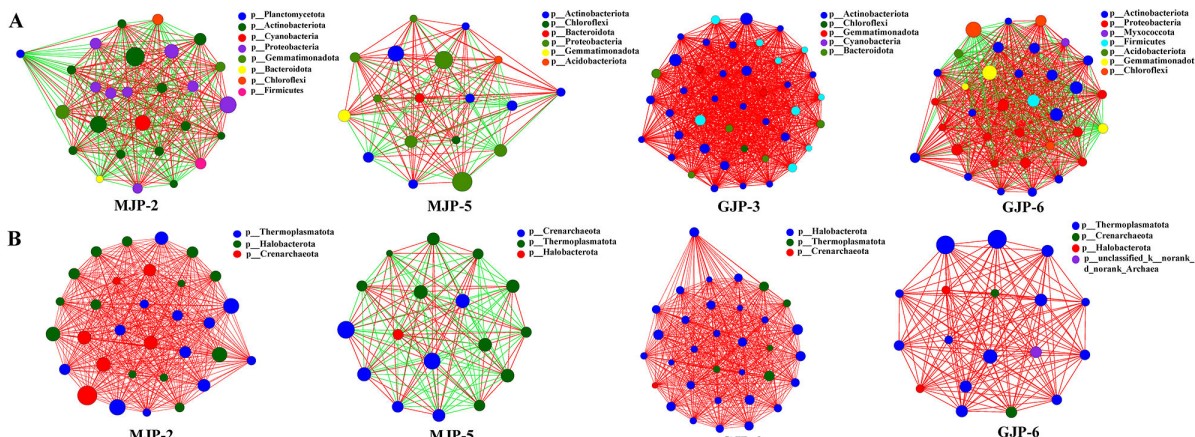

**FIG 7** Co-occurring analyze network of OTUs belonging to bacteria (A) and archaea (B) of *Suaeda dendroides* and *Suaeda altissima* rhizosphere. The top 50 OTUs of total abundance were selected. Each network node (individual circle) represents an OTU. The size of each node represents the abundance of species, the nodes are colored at the phylum level. The Spearman rank correlation coefficients were calculated to reflect the correlation between OTUs (r>|0.6|, *P* < 0.05). A red edge represents a positive interaction, and a green edge represents a negative interaction. The thickness of the line is proportional to the correlation coefficient between OTUs. The greater the number of lines indicates the more closely related that OTU is to the others.

and community structure of halophyte-associated rhizosphere microbes depend on the soil properties and plant species (38). Our research highlights the importance of understanding the complex interplay between soil properties, plant species, and microbial communities in saline soils. By gaining a deeper understanding of these relationships, we can develop more effective strategies for managing saline soils to enhance agricultural productivity.

## Soil salinity and *Suaeda* species co-shaped the specific structure of bacterial and archaeal community

Soil salinity can reduce soil microbial biomass, respiration, enzyme activity, and diversity (39). In our study, we observed that the bacterial and archaeal α-diversity index in the rhizosphere soil of *Suaeda* significantly reduced with the salinity increasing. The possible explanation for this negative effect could be attributed to the fact that the accumulation of salt in soils elevates the extracellular osmolarity, and many microorganisms that fail to adapt to osmotic stress may die or become inactive, thus reducing microbial alpha diversity (39). In addition, the increasing of soil saline would, therefore, mobilize soil organic carbon and other nutrients, which would support microbial growth. Such growth would result in higher microbial abundance but concomitantly decrease microbial diversity due to enrichment of a few community members (40). Previous studies have shown that besides soil salinity, pH is also a factor affecting the formation of bacterial and archaeal communities in halophytes rhizosphere (41, 42). This is consisted with our results; we found that soil pH, $Na^+$, EC have a strong correlation with the diversity and richness of microbial communities (Table S4B). Effect of soil pH was probably owing to the narrow pH ranges for optimal growth of soil bacteria and the role of soil pH in controlling accessibility of organic C and other nutrients (43). In addition, soil pH effected bacterial communities indirectly through quality and quantity of root exudates and ability in the release, uptake, and allocation of organic acids (44, 45). It can be seen that not only related to soil characteristics, but also plant physiological activity can influence the microbial community structure. Plant species have been proved that it can affect the structural and functional diversity of rhizosphere microbial community due to variations in root exudation and rhizodeposition in different zones (38, 46). In our study, we also confirmed this viewpoint and found significant differences in the structure of bacterial and archaeal community structures in the rhizosphere between *S. altissima* and *S. dendroides* (Fig. 3 and 4). To better adapt to the extreme hypersaline environment, halophytes could specifically recruit some plant beneficial bacterial taxa, such as nitrogen-fixing bacteria and extremely halophilic or halotolerant bacteria, to help them robustly grow and proliferate (47). We speculate that soil salinity and plant activity directly or indirectly regulate the structure and diversity of soil microbial communities in halophytes rhizosphere through different pathways.

## Increasing soil salinity reduced the complexity of the co-occurrence network

Co-occurrence network analysis has been commonly used to decipher the ecological relationships between microbial populations in diverse ecosystems. Our study investigates the effects of different salinity levels on the number of nodes and center coefficient in both bacteria and archaea networks. Notably, we found that salinity significantly reduced the complexity of microbial network, as observed in the rhizosphere of *Suaeda*. Our research are consistent with Zhao's findings; they indicated that nutrient-rich environments support more complex networks, while high salinity reduced microbial network complexity due to physical constraints such as increased soil bulk density, destroyed soil structure, and decreased porosity and oxygen concentration (48, 49). These factors can lead to a reduction in the effectiveness of soil nutrients and, thus, harmful consequences for the development of bacterial interaction. Furthermore, increasing salinity levels represent an abiotic stressor for bacteria and archaea that can filter out some low-adapted microbes and further reduce the complexity of microbial interaction networks. Our hypothesis is that *Suaeda* has simplified its rhizosphere

microbiome through a process of selection and assembly, favoring microbial communities with stronger stress tolerance, the ability to dominate ecological niches, and perform critical ecological function, thus eliminating a large number of microbial populations that do not have competitive advantages and leading to a reduction in the complexity of rhizosphere microbial population. The selection and assembly mechanism of *Suaeda* to their rhizosphere microbiomes under different environmental stresses remain to be further investigated.

## Unearthed of keystone taxa in the rhizosphere of *Suaeda*

We found that Actinobacteriota and Proteobacteria were the domain phyla in all samples. Previous research has shown that these bacteria are prevalent in soils affected by salinity and drought, as well as in the rhizosphere or endophytic flora of halophytes such as *Salcomia rubra*, *Leymus chinensis*, *Suaeda glauca*, and *Glaux maritima* (42, 50, 51). Actinobacteriota have been found to promote plant growth by producing antimicrobial compounds and degrading toxic organic compounds in polluted saline environments (52, 53). This suggests that Actinobacteriota may also play a crucial ecological role in the rhizosphere of *Suaeda* as a key microorganism. While Actinobacteria and Proteobacteria are well-known dominant bacterial taxa in soils and are known to contribute to plant growth and health, this study focuses on specific bacterial groups identified as keystone taxa in saline soils. These keystone taxa have the potential to play critical functional roles, and we are particularly interested in exploring their potential functions in promoting plant growth and health in these challenging environments.

We identified three important bacterial OTUs: OTU374 (*Halomonas*), OTU35 (*Arthroactor*), and OTU1757 (*Marmoricola*) by co-occurrence network analysis, which had high network center coefficient and played crucial roles in the soil's microbial network. We discovered that *Halomonas* was the most prevalent bacteria genus across all samples, with particularly high levels in MJP-5 and GJP-6. This suggests that *Halomonas* has remarkable adaptability and tolerance to high-salinity environments. Some research reported that *Halomonas elongate* is a type of salt-tolerant plant growth promoting rhizobium (ST-PGPR) that can enhance salt tolerance in alfalfa and purple basil by activating specific genes involved in photosynthesis, ion transport, plant hormones, and osmolyte production, increasing osmolytes, antioxidant enzymes activities (54, 55). Based on these findings, we hypothesize that *Halomonas* may also play an crucial ecological role in enhancing the salt tolerance and growth of *Suaeda* rhizosphere. *Arthrobacter* as an excellent PGPR has been reported to possess strong salt stress tolerance and giving it a competitive advantage in salt-stressed environments and an important niche (56). Some research reported that inoculating *Arthrobacter* improved pistachio and wheat seedlings growth, physiological and photosynthetic parameters, and resistance to salinity and drought stresses through regulating the expression of genes related to the production of auxin, ACC deaminase, siderophore, exopolysaccharides along with P/Zn solubilization activities (57, 58). In addition, we found that the abundance of *Marmoricola* particularly enriched in the high salinity soil (MJP-2 and GJP-3). They are a member of the Nocardioidaceae, which is widely present in nature enverionment (59). Li isolated *Marmoricola mangrovicus*, as an endophytic actinobacterium from *Kandelia candel*, and identified its catalase activity (60). In summary, we hypothesized these keystone taxa enhanced the salt tolerance and promoted plant growth of *Suaeda* by reducing reactive oxygen content, competing with other microorganisms for the ecology niche, and increasing the content of nutrients in the rhizosphere to support plant acquisition and absorption. OTU91 (*Candidatus_Nitrocosmicus*) as key archaea in the rhizosphere soil of *S. altissima* and *S. dendroides* belongs to a member of AOA that is involved in nitrification, the process of converting ammonium ($NH_4^+$) to nitrate ($NO_3^-$) (61). They are crucial for the nitrogen biogeochemical cycle due to participated in ecological functions associated with chemoheterotrophy, phototrophy, and aerobic ammonia oxidation (62). We hypothesize that salt stress reduced the nutrient content in soil, but the nutrient content increased of the plant rhizosphere

through ammonia oxidation activity of *Candidatus_Nitrocosmicus* that was benefited for plant growth and resistance of salt stress. Our results indicate that the rhizosphere of *Suaeda*, as an important resource bank of PGPR, playing an important ecological role in improving plant stress resistance and promoting plant growth.

## Conclusions

We investigated the effects of high and severe salinity soil on the bacterial and archaeal communities in the rhizosphere of *S. altissima* and *S. dendroides*. Our study showed that soil salinity and *Suaeda* species co-shaped community structure of bacteria and archaea. In addition, the complexity and diversity of co-occurrence network patterns of bacteria and archaea decreased with increasing soil salinity. We identified *Halomonas*, *Arthrobacter*, *Marmoricola,* and *Candidatus Nitrocosmicus* as key microorganisms in the *Suaeda* rhizosphere that played a vital role in promoting nitrogen cycling, improving plant salinity tolerance, and promoting plant growth. These findings have important implications for the spatial distribution and ecological diversity of bacteria and archaea in *S. altissima* and *S. dendroides* rhizosphere. Our results also provided a basis for development of beneficial microbial resources and ecological restoration in salinity-affected areas.

## MATERIALS AND METHODS

### Sample collection

The study was conducted in a saline field nearby Shihezi city in Xinjiang Province, China, which has a temperate continental climate with a mean air temperature of 10.15°C. Annual precipitation ranges from 1.16 to 28.76 mm, while annual evaporation ranges from 3.27 to 37.63 mm. The relative humidity is 46.91%, and the annual average ground-pressure is 965.17 hPa (data from NOAA-Climate Prediction Center, https://www.cpc.ncep.noaa.gov/; Global Modeling and Assimilation Office). The research involved collecting 12 rhizosphere soil samples from 4 locations in August 2018 (Table S2). The samples were obtained from the rhizosphere of naturally growing *S. dendroides* and *S. altissima*, which are domain halophyte species that grow in salinized soil. *S. dendroides* was found in locations A and C (marked MJP2 and MJP5, respectively), while *S. altissima* was found in locations B and D (marked GJP3 and GJP6, respectively). At each location, three sampling points were selected, each about 20 m apart, and three *Suaeda* plants with same growth status were obtained at each sampling point. We collected rhizosphere soil from *S. dendroides* and *S. altissima* by removing the surface soil 5 cm around the roots of the plant (Fig. 8). The roots were then dug out, and the tightly attached soil was shaken into sterile ziplock bags, labeled MJP2-1, MJP2-2, MJP2-3, GJP3-1, GJP3-2, GJP3-3, MJP5-1, MJP5-2, MJP5-3, GJP6-1, GJP6-2, and GJP6-3. The soil attached to the *Suaeda* root surface was gently shaken into a sterile plastic-sealing bag and stored on ice until transferred to the laboratory, and then, they were stored at −80°C for high-throughput sequencing. While the soil loosely attached root, after passing a 2-mm sieve, they were air-dried for analyzed soil physiological and biochemical parameters. We measured various environmental variables, including soil organic matters (OM), total carbon (TC), total nitrogen (TN), total phosphorus (TP), and available potassium (AK), $NH^{3+}$-N (AHN), available phosphorus (AP), available potassium (AK), electrical conductivity (EC), $Cl^-$, $SO_4^{2-}$, $Ca^{2+}$, $K^+$, $Mg^{2+}$, $Na^+$, $HCO_3^-$, microbial biomass carbon (MBC), catalase (CAT), urease (URA), protease (PRA), phosphatase (PHO), sucrase (SUC), and nitrite reductase (NIT) were measured as described by Bao (63).

### DNA extraction and PCR amplification, quantification, and purification

Microbial DNA was extracted from 1.0 g rhizosphere soil using the E.Z.N.A. soil DNA Kit (Omega Bio-tek, Norcross, GA, USA). The V4 hypervariable region of the bacterial and archaeal 16S rRNA gene were amplified using primer pairs (515F:

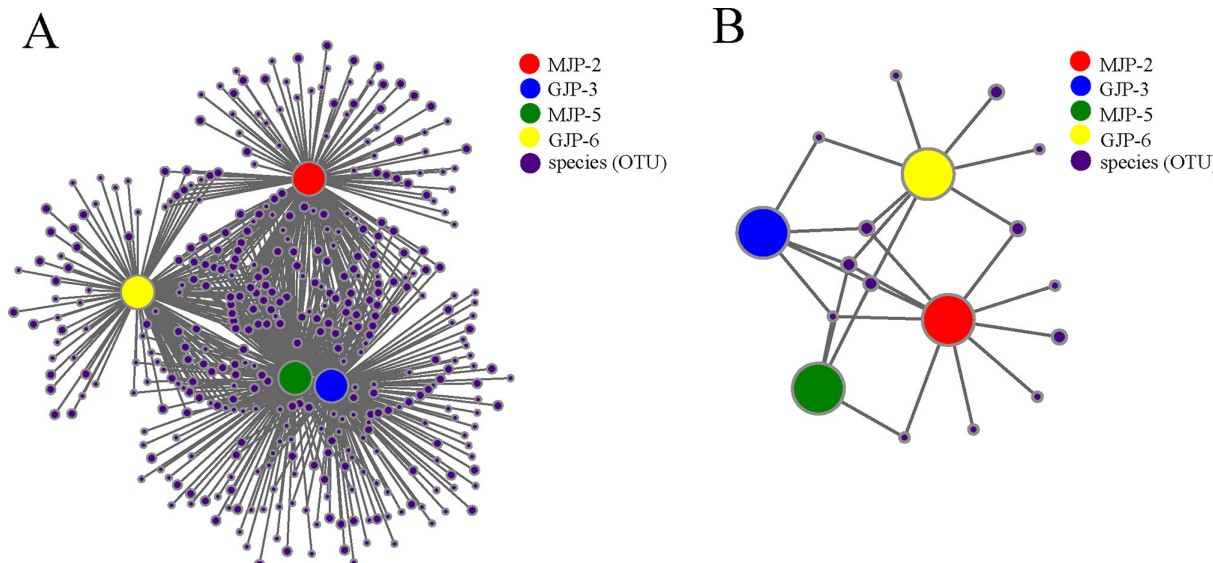

**FIG 8** The co-occurrence relationship network of bacterial (A) and archaeal (B) OTUs from 4 rhizosphere soil samples. The OTUs abundance (number of sequences) >50 was selected. Each network node (individual circle) represents an OTU. The size of each node represents the abundance of OTUs. The connection between sample nodes and OTU nodes represents that the sample contains the OTU.

3′-GTGCCAGCMGCCGCGG-5′；806R: 3′-GGACTACHVGGGTWTCTAAT-5′), with a unique barcode sequence for each sample by an ABI GeneAmp 9700 PCR thermocycler (ABI, CA, USA). The PCR mixture consisted of 4 µL 5× Fast Pfu buffer, 2 µL 2.5 mM dNTPs, 0.8 µL each primer (5 µM), 0.4 µL Fast Pfu polymerase, 10 ng of template DNA, and ddH$_2$O with a final volume of 20 µL. The PCR amplification cycling conditions were as follows: initial denaturation at 95℃ for 3 min, followed by 29 cycles of denaturing at 95℃ for 30 s, annealing at 53℃ for 30 s, and extension at 72℃ for 45 s, and single extension at 72℃ for 10 min, and end at 4℃. All samples were amplified in triplicate, and the PCR product was extracted from 2% agarose gel and purified using the AxyPrep DNA Gel Extraction Kit (Axygen Biosciences, Union City, CA, USA), following the manufacturer's instructions. The purified DNA was quantified using a Quantus Fluorometer (Promega, USA).

The equimolarly pooled, purified amplicons were subjected to paired-end sequencing on an Illumina Miseq PE300 platform by Majorbio Bio-Pharm Technology Co. Ltd. (Shanghai, China), following standard protocols by Illumina (San Diego, USA).

## Library preparation and sequencing

To prepare the libraries, we used "Y" adapters to link the PCR products. We removed adapters dimer using beads and PCR amplification to concentrate the libraries. Single-stranded DNA fragments were generated using sodium hydroxide. To sequence the samples, we pooled the libraries equimolarly and performed paired-end sequencing (2 × 250/300 bp) on an Illumina Miseq platform following standard protocols.

## Processing of sequencing data

In this study, we used FLASH 1.2.11software (64) to merge paired-end reads from DNA fragments and separated the resulting 16S rRNA gene sequences by sample based on their unique barcodes, allowing up to one mismatch. QIIME1.9.1 We filtered and removed reads that were too short, contained ambiguous bases, or could not be assembled using QIIME1.9.1 software. Then, we used UPARSE 7.1 to cluster the optimized sequences into operational taxonomic units (OTUs) using UPARSE 7.1 with 97% sequence similarity (65). To select representative sequence for each OTU, we chose the most abundant sequence. We also rarefied the number of 16S rRNA gene sequences from each sample to 20,000 to minimize the effects of sequencing depth on alpha and beta diversity measures. This

resulted in an average Good's coverage of 99.09%. We analyzed the taxonomy of each OTU representative sequence using RDP Classifier version 2.2 against the 16S rRNA gene database (Silva v.138) (https://www.arb-silva.de/) with a confidence threshold of 0.7 (66).

## Statistical analysis

The soil microbiota was analyzed using the Majorbio Cloud platform (https://cloud.majorbio.com). Mothur v1.30.1 (67) was used to calculate rarefaction curves, as well as alpha diversity indices such as observed OTUs (67). R vegan package (version 3.3.1) was used to create rank-abundance curves of bacterial and archaeal OTU.

The Chao1 and ACE indices were used to assess microbial community richness, while the Shannon and Simpson indices were used to analyze the microbial community diversity. QIIME 1.8.0 software was used to perform these calculations. Taxonomic analysis provided insight into the community structure of different samples. R vegan package (version 3.3.1) was used to create bacterial and archaeal community bar plots and heatmaps (68). The significance of environmental characteristics was determined using a one-way analysis of variance (ANOVA) test, performed using statistical package for the SPSS v.17.0. Principal component analysis (PCA) can demonstrate the similarity of microbial community structure among different samples. PCA uses an orthogonal transformation to convert a set of observations of possibly correlated variables into a set of values of linearly uncorrelated variables called principal components. We used the Kruskal-Wallis H test to compare the species different among all soil samples. To investigate the relationship between microbial community structure and environmental variables, we performed redundancy analysis (RDA) using the RStudio (v4.0.3) with the packages vegan v2.5.6 (69). We used forward selection based on Monte Carlo permutation tests (permutations = 9,999). The $x$- and $y$-axes, and the length of the corresponding arrows, represented the importance of each soil property in explaining the distribution of taxon across communities. We constructed co-occurrence networks in OTU level to explore the internal community relationships across the samples by Networkx software (70). Only Spearman correlations with an $r > 0.6$ ($P < 0.05$) were considered to indicate a valid interactive event. The size of nodes represents the abundance of OUT. The different colors of nodes represent the OTUs from different phylum. Red line represents positive correlation between different OTUs, and green represents negative correlation. Thicker line indicates there were higher correlation between OTUs. More lines connected with one OTU represent more closely related to other OTUs.

## ACKNOWLEDGMENTS

This work was supported by the National Key Technologies R&D Program of China (grant no. 2016YFC0501404). The funding agency has no role in the design, data collection, analysis or interpretation of the research or in the writing of the manuscript.

The authors would like to thank all the reviewers who participated in the review for their suggestions on this manuscript and for their help with the writing.

Q.W. designed and performed the experiments, analyzed the data, and drafted the manuscript. D.H. and X.Z. helped with data analysis, and revised and refined the manuscript. Y.C. collected the samples and analyzed part of the data. Y.S. and J.Z. designed and performed the experiments and analyzed the data. All authors read and approved the final version of the manuscript.

The authors declare that the research was conducted in the absence of any commercial or financial relationships that could be construed as a potential conflict of interest.

## AUTHOR AFFILIATION

[1]College of Life Sciences/Xinjiang Production and Construction Corps Key Laboratory of Oasis Town and Mountain-basin System Ecology, Shihezi University, Shihezi, Xinjiang, China

## AUTHOR ORCIDs

Qiqi Wang  http://orcid.org/0000-0002-6588-8763
Yanfei Sun  http://orcid.org/0000-0003-1947-9245
Jianbo Zhu  http://orcid.org/0000-0002-6622-9014

## FUNDING

| Funder | Grant(s) | Author(s) |
| --- | --- | --- |
| The National Key Technologies R&D Program of China | 2016YFC0501404 | Yanfei Sun |

## AUTHOR CONTRIBUTIONS

Qiqi Wang, Data curation, Methodology, Writing – original draft | Dalun He, Data curation, Methodology, Resources, Software | Xinrui Zhang, Data curation, Methodology, Software, Writing – review and editing | Yongxiang Cheng, Investigation, Resources, Software, Writing – review and editing | Yanfei Sun, Investigation, Resources, Writing – review and editing | Jianbo Zhu, Resources, Writing – review and editing

## DATA AVAILABILITY

Sequence data of this project have been deposited in the Sequence Read Archive (SRA) of the National Center for Biotechnology Information (NCBI) under the accession number: PRJNA762470.

## ADDITIONAL FILES

The following material is available online.

### Supplemental Material

**Fig. S1 (Spectrum01649-23-S0001.tif).** Rarefaction curves based on the sequences of the V4 region of the 16S rRNA gene from samples associated with rhizosphere soil samples from *Suaeda dendroides* and *Suaeda altissima*.
**Fig. S2 (Spectrum01649-23-S0002.tif).** Rank-abundance curves on OTU level from samples associated with rhizosphere soil samples from *Suaeda dendroides* and *Suaeda altissima*.
**Supplemental legends (Spectrum01649-23-S0003.docx).** Legends for supplemental tables and figures.
**Supplemental tables (Spectrum01649-23-S0004.docx).** Tables S1 to S4.
**Table S5 (Spectrum01649-23-S0005.xlsx).** Node properties and centrality coefficients of the bacterial covariance correlation network in various samples.
**Table S6 (Spectrum01649-23-S0006.xlsx).** Node properties and center coefficients of bacterial and archaeal co-occurrence networks in all samples.

### Open Peer Review

**PEER REVIEW HISTORY (review-history.pdf).** An accounting of the reviewer comments and feedback.

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
