## [Reviewer comments · Microbiology Spectrum]

Microbiology Spectrum

Insight into bacterial and archaeal community structure of *Suaeda altissima* and *Suaeda dendroides* rhizosphere in response to different salinity level

qiqi Wang, Dalun He, Xinrui Zhang, Yongxiang Cheng, Yan Sun, and Jianbo Zhu

Corresponding Author(s): Yan Sun, Shihezi University

Review Timeline:

Submission Date:	April 21, 2023
Editorial Decision:	June 5, 2023
Revision Received:	August 21, 2023
Editorial Decision:	August 30, 2023
Revision Received:	September 5, 2023
Accepted:	September 8, 2023

Editor: Jing Han

Reviewer(s): The reviewers have opted to remain anonymous.

Transaction Report:

DOI: <https://doi.org/10.1128/spectrum.01649-23>

June 5, 2023

Prof. Yanfei Sun
Shihezi University
Shihezi
China

Re: Spectrum01649-23 (Analysis of bacterial and archaeal community structure in *Suaeda altissima* and *Suaeda dendroides* rhizosphere under different salinity level)

Dear Prof. Yanfei Sun:

Link Not Available

Sincerely,

Jing Han

Journals Department
Reviewer comments:

Reviewer #1 (Public repository details (Required)):

The sequencing data, which has been deposited in NCBI by authors.

Reviewer #1 (Comments for the Author):

The study from Wang et al. investigated the bacterial and archaeal community composition and diversity in the two types of *Suaeda* rhizosphere along different salinity gradient. The authors focused on how soil properties regulated the bacterial and archaeal communities diversity and networks, which is interesting. However, there are a number of issues that the authors need to address and revise carefully, as well as more in-deep discussion needs to be added for a better elucidation of the results.

Moreover, the results and discussion section were not well organized, and there are many writing errors in the manuscript and all citation formats are incorrect. I provide the authors with specific and textual comments below, I hope they find them useful in the further improvement of their manuscript.

General comments:

1. Title: The current version is too normal and is not attractive. Please revise it appropriately.
2. Since this study aims to investigate the variations in the rhizosphere microbial community of two types of Suaeda plants under different levels of soil salinity, it is important for the authors to provide a clear introduction to the differences in how these plants respond to soil salinization. How these two Suaeda types response to salt stress? Why they need to investigate the rhizosphere microbial communities? Were there any important research gaps/hypotheses that need to be elucidated so that make the mechanisms behind more clear? It may be difficult to understand the observed differences in microbial communities between the two plant types and the purpose of this study if lack these information. Therefore, the authors should provide a thorough explanation of these two plant responses to soil salinity to support their findings.
3. The archaeal community is extracted from 16S amplicon sequencing data using bacteria-specific primers, thus there are very few archaea detected using this method. This is important and need to be discussed carefully. Moreover, I did not find any results related to functional prediction in the study, although the authors mentioned it in the M&M section. Therefore, it is unclear whether functional prediction was actually performed and, if so, what the results were. The authors should provide more clarity on this matter and clearly state whether functional prediction was performed and include the results in their manuscript.
4. As mentioned in results, soil pH and salinity level are all key factor influencing microbial community assembly, however, why the authors declare that "soil salinity and Suaeda species co-shaped bacterial and archaeal community structure"? There is no bulk soil samples here. More direct proof should be provided.
5. The discussion and conclusion sections of the manuscript are somewhat vague and lack clear conclusions, especially in the discussion section, the authors display many descriptive expressions and repeat their results, which does not provide a satisfactory analysis of their findings. Therefore, it is recommended that the discussion and conclusion be rewritten with a more critical and concise method that focuses on interpreting the results in light of the research questions and objectives. This will help to provide a clearer and more convincing conclusion and make the manuscript higher quality.
6. Grammar and writing: The writing of the text needs to be carefully checked. So many incorrect wordings/phrases, font format, punctuation mark, and citation format. Please find more specific comments below.

Abstract:

General comments:

I believe the main issue with this part is the language accuracy and conciseness. The authors used many short sentences that lack strong connections between each other, and the Abstract is not concise. To improve the readability and logicity of the text, the authors should consider adding conjunctions and reducing descriptive statements. Although I have pointed out some places, there are still many areas that need to be revised. Additionally, it would be beneficial to add some clear conclusions at the end of the Abstract.

Specific comments:

7. Line 19. Which kind of agricultural soil? It is better to emphasize "the improvement of saline soil".
8. Line 19-20. Reword the sentence.
9. Line 20. However, ...
10. Line 24. "strongly and very strongly..." Please reword this sentence. If possible, the authors can add the specific level of salinity in two soil types to provide a more detailed understanding of the impact of salinity on the soil microbial community.
11. Line 27 & 29, Its all *S. altissima*?
12. Line 31. "showed declined". Reword this sentence, it's too chinglish.
13. Line 35-36. Why soil salinity influence rhizosphere microbial community? Based on the previous results, soil pH was the main factor. Please explain this.
14. Line 46. ... important ecological roles.
15. Line 45-46. Reword the sentence.

Introduction:

General comments:

Since this study introduce the importance of bacterial and archaeal community of two Suaeda types under different salt stress and aimed to analyze the variations in community diversity and composition, the authors should clearly introduce the effects of soil salinity on rhizosphere microbiomes. How the rhizosphere bacterial and archaeal community influence the plant fitness to soil salinization? Why they need to be investigated? What's the difference between halophytes and glycophytes rhizosphere microbiomes? Were there any important research gaps/hypotheses that need to be elucidated so that make the microbial mechanisms behind more clear?

Specific comments:

1. Line 55. Whose functions?
2. Line 62-64. Suggest rewording this statement.
3. Line 74. ... in agricultural management.
4. Line 94. Adding "Therefore" before "Suaeda's ability...".
5. Line 97. Providing some references to support your statements.
6. Line 98. Plant growth promoting rhizobacteria (PGPR)...
7. Line 101. Dose *Sphingobacterium* belong to PGPR? Suggest rewording this sentence to avoid ambiguous expression.
8. Line 108-109. Reword.
9. Line 114. Tan et al.,

10. Line 119-120. ..., and it is necessary to ...
11. Lines 97-118. The authors should add some statements about the importance of archaea in plants resistance on soil salinity.
12. Line 122. ... strong and very strongly salinity levels? Please change the expression.

Materials and methods:

1. Line 141. *S. altissima* was found in B and D according to Table S2?
2. If possible, I suggest using figures to clearly express your field experiment and sampling strategy.
3. Line 148-150. Suggest rewording this sentence.
4. Line 202. The authors predict the functions of bacterial community using Tax4Fun. However, I did not see any relevant results about microbial functions in the manuscript. It is inappropriate and more direct analyses should be provided.
5. Line 217-219. Delete this sentence.
6. Line 212 and 225, vegan package needs to be cited.
7. Line 228-230. How you build the networks? What package you used? How the networks visualized? Please add the detailed expression.

Results:

1. All the figures are fuzzy, the font size is too small. The authors should revise to make the figures clearer.
2. Line 239. ... soil samples from other three locations.
3. Line 251. It is better to change "the soil of GIP-3 has..." as "soil samples from GIP-3 have...".
4. Line 270. Please add the cited Figure or Table to support your statement.
5. Line 346-348. Add the related Figures or Tables.
6. Line 388. OUTs? correlations.
7. Line 393. The network edges is degree?
8. Line 396. How you define the complexity of microbial networks? The number of edges and nodes? Or some other parameters?
9. Line 400. Cyanobacteria et. al.? Is it correct?
10. Line 414. Where is the Table 5B???

Discussion:

Discussion section is too lengthy and need to be largely revised. The authors should delete many descriptive expressions. Most importantly, authors need to strengthen the logicity of the discussion. Additionally, the discussion section does not require so many sub-titles, short and concise titles should be summarized in this section.

1. Line 427. microbes
2. Line 428. various species? There are only two different types here.
3. Line 448. Reword this sentence.
4. Lines 449-452. Why did the authors get this conclusion? Add some related results and references to enhance your statements.
5. Lines 452-454. The authors need to elucidate why and how the previous studies' findings consistent with your results.
6. Line 465. What is the purpose of introducing "core microbiome" here? Dose it relevant to your results?
7. Line 474. Why did the authors discuss the pathogen suppressive of Actinobacteria? I did not see any results about the pathogens in your results. Thus, I suggest to carefully reconsider the logicity of these sentences.
8. Line 477-478. While Actinobacteria and Proteobacteria are well-known dominant bacterial taxa in soils and are known to contribute to plant growth and health, this study focuses on specific bacterial groups identified as keystone taxa in saline soils. These keystone taxa have the potential to play critical functional roles, and we are particularly interested in exploring their potential functions in promoting plant growth and health in these challenging environments.
9. Line 501-502. should include the relevant results and discuss more about this taxa to explain the importance.
10. Line 508. "*Marmoricola*" should be italic.
11. Line 519. They are often found in...
12. Line 522-526. Reword this sentence.
13. Line 529-530. Cited the relevant reference(s).
14. Line 552-554. It's hard to understand the purpose of this paragraph and its connection with above.
15. Line 557-568. Soil pH is an important factor that driving soil microbial community assembly. However, in this study, authors should focus on the soil salinization.
16. Line 574. What is "t microbial network"?
17. Line 579-580. What is the relationship between theses physical constraints and microbial networks? These sentences are vague and need to be reword.
18. Lines 584-590. The authors' hypotheses require further elaboration and deeper discussion. It is unclear how they arrived at these hypotheses, and more statements and evidence are needed to support their claims.
19. Line 595. soil's network is vague.

Staff Comments:

Preparing Revision Guidelines

To submit your modified manuscript, log onto the eJP submission site at <https://spectrum.msubmit.net/cgi-bin/main.plex>. Go to

Author Tasks and click the appropriate manuscript title to begin the revision process. The information that you entered when you first submitted the paper will be displayed. Please update the information as necessary. Here are a few examples of required updates that authors must address:

Please return the manuscript within 60 days; if you cannot complete the modification within this time period, please contact me. If you do not wish to modify the manuscript and prefer to submit it to another journal, please notify me of your decision immediately so that the manuscript may be formally withdrawn from consideration by Microbiology Spectrum.

The study from Wang et al. investigated the bacterial and archaeal community composition and diversity in the two types of *Suaeda* rhizosphere along different salinity gradient. The authors focused on how soil properties regulated the bacterial and archaeal communities diversity and networks, which is interesting. However, there are a number of issues that the authors need to address and revise carefully, as well as more in-deep discussion needs to be added for a better elucidation of the results. Moreover, the results and discussion section were not well organized, and there are many writing errors in the manuscript and all citation formats are incorrect. I provide the authors with specific and textual comments below, I hope they find them useful in the further improvement of their manuscript.

General comments:

1. Title: The current version is too normal and is not attractive. Please revise it appropriately.
2. Since this study aims to investigate the variations in the rhizosphere microbial community of two types of *Suaeda* plants under different levels of soil salinity, it is important for the authors to provide a clear introduction to the differences in how these plants respond to soil salinization. How these two *Suaeda* types response to salt stress? Why they need to investigate the rhizosphere microbial communities? Were there any important research gaps/hypotheses that need to be elucidated so that make the mechanisms behind more clear? It may be difficult to understand the observed differences in microbial communities between the two plant types and the purpose of this study if lack these information. Therefore, the authors should provide a thorough explanation of these two plant responses to soil salinity to support their findings.
3. The archaeal community is extracted from 16S amplicon sequencing data using bacteria-specific primers, thus there are very few archaea detected using this method. This is important and need to be discussed carefully. Moreover, I did not find any results related to functional prediction in the study, although the authors

mentioned it in the M&M section. Therefore, it is unclear whether functional prediction was actually performed and, if so, what the results were. The authors should provide more clarity on this matter and clearly state whether functional prediction was performed and include the results in their manuscript.

4. As mentioned in results, soil pH and salinity level are all key factors influencing microbial community assembly, however, why the authors declare that “soil salinity and *Suaeda* species co-shaped bacterial and archaeal community structure”? There is no bulk soil samples here. More direct proof should be provided.
5. The discussion and conclusion sections of the manuscript are somewhat vague and lack clear conclusions, especially in the discussion section, the authors display many descriptive expressions and repeat their results, which does not provide a satisfactory analysis of their findings. Therefore, it is recommended that the discussion and conclusion be rewritten with a more critical and concise method that focuses on interpreting the results in light of the research questions and objectives. This will help to provide a clearer and more convincing conclusion and make the manuscript higher quality.
6. Grammar and writing: The writing of the text needs to be carefully checked. So many incorrect wordings/phrases, font format, punctuation mark, and citation format. Please find more specific comments below.

Abstract:

General comments:

I believe the main issue with this part is the language accuracy and conciseness. The authors used many short sentences that lack strong connections between each other, and the Abstract is not concise. To improve the readability and logic of the text, the authors should consider adding conjunctions and reducing descriptive statements. Although I have pointed out some places, there are still many areas that need to be

revised. Additionally, it would be beneficial to add some clear conclusions at the end of the Abstract.

Specific comments:

7. Line 19. Which kind of agricultural soil? It is better to emphasize “the improvement of saline soil”.
8. Line 19-20. Reword the sentence.
9. Line 20. However, ...
10. Line 24. “strongly and very strongly...” Please reword this sentence. If possible, the authors can add the specific level of salinity in two soil types to provide a more detailed understanding of the impact of salinity on the soil microbial community.
11. Line 27 & 29, Its all *S. altissima*?
12. Line 31. “showed declined”. Reword this sentence, it’s too chinglish.
13. Line 35-36. Why soil salinity influence rhizosphere microbial community? Based on the previous results, soil pH was the main factor. Please explain this.
14. Line 46. ... important ecological roles.
15. Line 45-46. Reword the sentence.

Introduction:

General comments:

Since this study introduce the importance of bacterial and archaeal community of two Suaeda types under different salt stress and aimed to analyze the variations in community diversity and composition, the authors should clearly introduce the effects of soil salinity on rhizosphere microbiomes. How the rhizosphere bacterial and archaeal community influence the plant fitness to soil salinization? Why they need to be investigated? What’s the difference between halophytes and glycophytes rhizosphere microbiomes? Were there any important research gaps/hypotheses that need to be elucidated so that make the microbial mechanisms behind more clear?

Specific comments:

1. Line 55. Whose functions?
2. Line 62-64. Suggest rewording this statement.
3. Line 74. ... in agricultural management.
4. Line 94. Adding “Therefore” before “Suaeda's ability...”.
5. Line 97. Providing some references to support your statements.
6. Line 98. Plant growth promoting rhizobacteria (PGPR)...
7. Line 101. Dose Sphingobacterium belong to PGPR? Suggest rewording this sentence to avoid ambiguous expression.
8. Line 108-109. Reword.
9. Line 114. Tan et al.,
10. Line 119-120. ..., and it is necessary to ...
11. Lines 97-118. The authors should add some statements about the importance of archaea in plants resistance on soil salinity.
12. Line 122. ... strong and very strongly salinity levels? Please change the expression.

Materials and methods:

1. Line 141. *S. altissima* was found in B and D according to Table S2?
2. If possible, I suggest using figures to clearly express your field experiment and sampling strategy.
3. Line 148-150. Suggest rewording this sentence.
4. Line 202. The authors predict the functions of bacterial community using Tax4Fun. However, I did not see any relevant results about microbial functions in the manuscript. It is inappropriate and more direct analyses should be provided.
5. Line 217-219. Delete this sentence.
6. Line 212 and 225, vegan package needs to be cited.
7. Line 228-230. How you build the networks? What package you used? How the networks visualized? Please add the detailed expression.

Results:

1. All the figures are fuzzy, the font size is too small. The authors should revise to make the figures clearer.
2. Line 239. ... soil samples from other three locations.
3. Line 251. It is better to change “the soil of GIP-3 has...” as “soil samples from GIP-3 have...”.
4. Line 270. Please add the cited Figure or Table to support your statement.
5. Line 346-348. Add the related Figures or Tables.
6. Line 388. OUTs? correlations.
7. Line 393. The network edges is degree?
8. Line 396. How you define the complexity of microbial networks? The number of edges and nodes? Or some other parameters?
9. Line 400. Cyanobacteria et. al.? Is it correct?
10. Line 414. Where is the Table 5B???

Discussion:

Discussion section is too lengthy and need to be largely revised. The authors should delete many descriptive expressions. Most importantly, authors need to strengthen the logicity of the discussion. Additionally, the discussion section does not require so many sub-titles, short and concise titles should be summarized in this section.

1. Line 427. microbes
2. Line428. various species? There are only two different types here.
3. Line 448. Reword this sentence.
4. Lines 449-452. Why did the authors get this conclusion? Add some related results and references to enhance your statements.
5. Lines 452-454. The authors need to elucidate why and how the previous studies' findings consistent with your results.

6. Line 465. What is the purpose of introducing “core microbiome” here? Dose it relevant to your results?
7. Line 474. Why did the authors discuss the pathogen suppressive of Actinobacteria? I did not see any results about the pathogens in your results. Thus, I suggest to carefully reconsider the logicity of these sentences.
8. Line 477-478. While Actinobacteria and Proteobacteria are well-known dominant bacterial taxa in soils and are known to contribute to plant growth and health, this study focuses on specific bacterial groups identified as keystone taxa in saline soils. These keystone taxa have the potential to play critical functional roles, and we are particularly interested in exploring their potential functions in promoting plant growth and health in these challenging environments.
9. Line 501-502. should include the relevant results and discuss more about this taxa to explain the importance.
10. Line 508. “Marmoricola” should be italic.
11. Line 519. They are often found in...
12. Line 522-526. Reword this sentence.
13. Line 529-530. Cited the relevant reference(s).
14. Line 552-554. It’s hard to understand the purpose of this paragraph and its connection with above.
15. Line 557-568. Soil pH is an important factor that driving soil microbial community assembly. However, in this study, authors should focus on the soil salinization.
16. Line 574. What is “t microbial network”?
17. Line 579-580. What is the relationship between theses physical constraints and microbial networks? These sentences are vague and need to be reword.
18. Lines 584-590. The authors' hypotheses require further elaboration and deeper discussion. It is unclear how they arrived at these hypotheses, and more statements and evidence are needed to support their claims.
19. Line 595. soil’s network is vague.

Dear reviewers 1:

Thank you for giving us the opportunity to submit a revised draft of the manuscript “Analysis of bacterial and archaeal community structure in *Suaeda altissima* and *Suaeda dendroides* rhizosphere under different salinity level” (Spectrum01649-23R1) for publication in *Microbiology Spectrum*. We appreciate the time and effort that you dedicated to providing feedback on our manuscript and are grateful for the insightful comments on and valuable improvements to our paper. We have incorporated the suggestions made by the reviewers. Please see below for a point-by-point response to the reviewers’ comments and concerns.

General comments:

1. Title: The current version is too normal and is not attractive. Please revise it appropriately.

Reply : Thanks for your suggestions, we have changed the title from “Analysis of bacterial and archaeal community structure in *Suaeda altissima* and *Suaeda dendroides* rhizosphere under different salinity level” to “Insight into bacterial and archaeal community structure of *Suaeda altissima* and *Suaeda dendroides* rhizosphere in response to different salinity level”.

2. Since this study aims to investigate the variations in the rhizosphere microbial community of two types of *Suaeda* plants under different levels of soil salinity, it is important for the authors to provide a clear introduction to the differences in how these plants respond to soil salinization. How these two *Suaeda* types response to salt stress? Why they need to investigate the rhizosphere microbial communities? Were there any important research gaps/hypotheses that need to be elucidated so that make the mechanisms behind more clear? It may be difficult to understand the observed differences in microbial communities between the two plant types and the purpose of this study if lack these informations. Therefore, the authors should provide a thorough explanation of these two plant responses to soil salinity to support their findings.

Reply: Thanks for your suggestions. We have added the *Suaeda* plants respond to soil salinization and the importance and reasons of rhizosphere microorganisms for plant salt tolerance and growth in introduction section.

3. The archaeal community is extracted from 16S amplicon sequencing data using bacteria-specific primers, thus there are very few archaea detected using this method. This is important and need to be discussed carefully. Moreover, I did not find any results related to functional prediction in the study, although the authors mentioned it in the M&M section. Therefore, it is unclear whether functional prediction was actually performed and, if so, what the results were. The authors should provide more clarity on this matter and clearly state whether functional prediction was performed and include the results in their manuscript.

Reply: Thanks for your suggestion. We referred to common primers for bacteria

and archaea involved in some studies. We accept your feedback and will use archaea specific primers for more accurate research in future studies. Thank you very much. For the functional prediction in the result, we did not conduct functional prediction analysis, but due to my negligence added this method, we have removed its description.

[1] Steffi, Genderjahn, Mashal, Alawi, Kai, & Mangelsdorf, et al. (2018). Desiccation- and saline-tolerant bacteria and archaea in kalahari pan sediments. *Front. Microbiol.* 9: 2082.

[2] Wang Y, Dang N, Feng K, Wang J, Jin X, Yao S, Wang L, Gu S, Zheng H, Lu G and Deng Y (2023) Grass-microbial inter-domain ecological networks associated with alpine grassland productivity. *Front. Microbiol.* 14:1109128.

4. As mentioned in results, soil pH and salinity level are all key factor influencing microbial community assembly, however, why the authors declare that “soil salinity and *Suaeda* species co-shaped bacterial and archaeal community structure”? There is no bulk soil samples here. More direct proof should be provided.

Reply: Thanks for your suggestions. We have modified this section.

5. The discussion and conclusion sections of the manuscript are somewhat vague and lack clear conclusions, especially in the discussion section, the authors display many descriptive expressions and repeat their results, which does not provide a satisfactory analysis of their findings. Therefore, it is recommended that the discussion and conclusion be rewritten with a more critical and concise method that focuses on interpreting the results in light of the research questions and objectives. This will help to provide a clearer and more convincing conclusion and make the manuscript higher quality.

Reply: Thanks for your suggestions. We've rewritten the discussion and conclusion sections.

6. Grammar and writing: The writing of the text needs to be carefully checked. So many incorrect wordings/phrases, font format, punctuation mark, and citation format. Please find more specific comments below.

Reply: Thanks for your suggestions. We have corrected the text for incorrect grammar and citation formatting according to specific comments below. We have tried our best to polish the language in the revised manuscript.

Abstract:

General comments:

I believe the main issue with this part is the language accuracy and conciseness. The authors used many short sentences that lack strong connections between each other, and the Abstract is not concise. To improve the readability and logicity of the text, the authors should consider adding conjunctions and reducing descriptive statements. Although I have pointed out some places, there are still many areas that need to be

revised. Additionally, it would be beneficial to add some clear conclusions at the end of the Abstract.

Specific comments:

Reply: Thanks for your suggestion. We have checked the language accuracy and conciseness. We have adding conjunctions and reducing descriptive statements. We made a big modify to the Abstract.

7. Line 19. Which kind of agricultural soil? It is better to emphasize “the improvement of saline soil”.

Reply: Thanks for your suggestion. We have emphasized “saline soil” in Abstract section.

8. Line 19-20. Reword the sentence.

Reply: Thanks for your suggestion. We have reworded the abstract.

9. Line 20. However, ...

Reply: Thanks for your suggestion. We have added the “However”.

10. Line 24. “strongly and very strongly...” Please reword this sentence. If possible, the authors can add the specific level of salinity in two soil types to provide a more detailed understanding of the impact of salinity on the soil microbial community.

Reply: Thanks for your suggestion. We have reworded this sentence. We have referred the expressions based on the Food and Agriculture Organization (FAO) soil salinity classification system, description as: 1) very strongly saline, >16 dS/m; (2) strongly saline, 8–16 dS/m; (3) moderately saline, 4–8 dS/m; (4) slightly saline, 2–4 dS/m; and (5) non-saline, 0–2 dS/m.”

We have added the specific level of salinity, “strongly (EC, 8-16 dS/m) and very strongly (EC, >16 dS/m) saline soil”.

11. Line 27 & 29, Its all *S. altissima*?

Reply: Thanks for your reminder. We have rewritten the Abstract section.

12. Line 31. “showed declined”. Reword this sentence, it’s too chinglish.

Reply: Thanks for your suggestion. We have reworded this sentence.

13. Line 35-36. Why soil salinity influence rhizosphere microbial community? Based on the previous results, soil pH was the main factor. Please explain this.

Reply: In our study, soil types were categorized as heavily saline and very heavily saline based on soil electrical conductivity. Soil salinity level is judged by the magnitude of soil EC, which includes all the ions and electrolytes in the soil. Under natural conditions, changes in soil salinity levels are accompanied by changes in soil pH, and these changes in pH are partly due to the presence of Na_2CO_3 and NaHCO_3 , which complement each other. We therefore wanted to

investigate the main factors that alter the composition and structure of plant rhizosphere microbial communities. Therefore, we measured some physicochemical parameters related to soil salinity, including soil pH. By Monte Carlo analysis and RDA analysis, we identified the most relevant factors on community composition, among which pH also was significantly correlated with bacterial archaeal community structure. In broad terms, the large differences of microbial community structure were caused by different salinity levels. In detail, it was due to changes in some ion contents in the soil and changes in pH that drove changes in plant inter-root microbial community structure.

14. Line 46. ... important ecological roles.

Reply: Thanks for your suggestion. We have changed “important implication” to “important ecological roles”.

15. Line 45-46. Reword the sentence.

Reply: We have reworded this sentence. Our study has important ecological roles for fully understanding soil salinization and halophyte species on the spatial distribution and ecological diversity of bacteria and archaea, and we provide a basis for biological improvement and ecological restoration of salinity-affected areas.

Introduction:

General comments:

Since this study introduce the importance of bacterial and archaeal community of two *Suaeda* types under different salt stress and aimed to analyze the variations in community diversity and composition, the authors should clearly introduce the effects of soil salinity on rhizosphere microbiomes. How the rhizosphere bacterial and archaeal community influence the plant fitness to soil salinization? Why they need to be investigated? What’s the difference between halophytes and glycophytes rhizosphere microbiomes? Were there any important research gaps/hypotheses that need to be elucidated so that make the microbial mechanisms behind more clear?

Reply: Thanks for your suggestion. We have rewritten the introduction section, added the interaction between the rhizosphere bacterial and archaeal community and the plant fitness to soil salinization, the reasons of them need to be investigated. We added the research hypotheses in the last graph of instruction section.

Specific comments:

1. Line 55. Whose functions?

Reply: we added “soil productivity” in this sentence.

2. Line 62-64. Suggest rewording this statement.

Reply: Thanks for your suggestion. We have rewritten this paragraph.

3. Line 74. ... in agricultural management.
Reply: We have added “in agricultural management.” in sentence.
4. Line 94. Adding “Therefore” before “*Suaeda's* ability...”.
Reply: Thanks for your suggestion. We have adding “Therefore” before “*Suaeda's* ability...”.
5. Line 97. Providing some references to support your statements.
Reply: Thanks for your suggestion. We have rewritten introduction section, and added some references to support my statements.
6. Line 98. Plant growth promoting rhizobacteria (PGPR)...
Reply: Thanks for your suggestion. we have changed the sentence.
7. Line 101. Dose *Sphingobacterium* belong to PGPR? Suggest rewording this sentence to avoid ambiguous expression.
Reply: Thanks for your suggestion. We have reworded this sentence
8. Line 108-109. Reword.
Reply: Thanks for your suggestion. We have reworded this sentence
9. Line 114. Tan et al.,
Reply: Thanks for your suggestion. We have rewritten this sentence.
10. Line 119-120. ..., and it is necessary to ...
Reply: Thanks for your suggestion. We have reworded this sentence.
11. Lines 97-118. The authors should add some statements about the importance of archaea in plants resistance on soil salinity.
Reply: Thanks for your suggestion. We have added some statements about the importance of archaea in plants resistance on soil salinity.
12. Line 122. ... strong and very strongly salinity levels? Please change the expression.
Reply: Thanks for your suggestion. I have referred to some paper expressions and changed the strong salinity level to high salinity level, very strong salinity level to sever salinity level.
“Five salinity classes were established using the EC values based on the Food and Agriculture Organization (FAO) soil salinity classification system as follows: 1) very strongly saline, >16 dS/m; (2) strongly saline, 8–16 dS/m; (3) moderately saline, 4–8 dS/m; (4) slightly saline, 4–2 dS/m; and (5) non-saline, 0–2 dS/m.” is displayed in the results section.
Aldakheel, Y., Y., Amal, Allbed, & Lalit, et al. (2014). Assessing soil salinity using soil salinity and vegetation indices derived from ikonos high-spatial resolution

imageries: applications in a date palm dominated region. *Geoderma: An International Journal of Soil Science*.

Abrol, I.P., Yadav, J.S.P., Massoud, F.I., 1988. *Salt-affected Soils and Their Management*. FAO.

Materials and methods:

1. Line 141. *S. altissima* was found in B and D according to Table S2?

Reply: Thanks for your suggestion. We have corrected this sentence.

2. If possible, I suggest using figures to clearly express your field experiment and sampling strategy.

Reply: Thanks for your suggestion. We added a sampling model figure named Figure 1.

3. Line 148-150. Suggest rewording this sentence.

Reply: Thanks for your suggestion. We have reworded this sentence.

4. Line 202. The authors predict the functions of bacterial community using Tax4Fun. However, I did not see any relevant results about microbial functions in the manuscript. It is inappropriate and more direct analyses should be provided.

Reply: Sorry. Due to our negligence, we wrote this part of the content in excess. We have deleted this sentence.

5. Line 217-219. Delete this sentence.

Reply: Thanks for your suggestion. We have deleted this sentence.

6. Line 212 and 225, vegan package needs to be cited.

Reply: Thanks for your suggestion. We have added the reference.

7. Line 228-230. How you build the networks? What package you used? How the networks visualized? Please add the detailed expression.

Reply: Thanks for your suggestion. We have added the detailed expression about the networks.

Results:

1. All the figures are fuzzy, the font size is too small. The authors should revise to make the figures clearer.

Reply: Thanks for your suggestion. We have adjusted the suitable size of the font and replaced clearer figures.

2. Line 239. ... soil samples from other three locations.

Reply: Thanks for your suggestion. We have added "soil samples from other three locations" in this sentence.

3. Line 251. It is better to change “the soil of GIP-3 has...” as “soil samples from GIP-3 have...”.

Reply: Thanks for your suggestion. We have changed “the soil of GIP-3 has...” as “soil samples from GIP-3 have...”.

4. Line 270. Please add the cited Figure or Table to support your statement.

Reply: Thanks for your suggestion. We have added the Supplement Figure to support our statement.

5. Line 346-348. Add the related Figures or Tables.

Reply: Thanks for your suggestion. We have added the Supplement figure 2 in this sentence.

6. Line 388. OUTs? correlations.

Reply: Thanks for your suggestion. We have corrected this error, and changed OUTs to OTUs.

7. Line 393. The network edges is degree?

Reply: The network edges is degree, they all represent the connecting line between species or between species and samples.

8. Line 396. How you define the complexity of microbial networks? The number of edges and nodes? Or some other parameters?

Reply: Thanks for your suggestion. The complexity of microbial network can be evaluated by the numbers of edges and nodes. We have added the number of edges and nodes.

9. Line 400. Cyanobacteria et. al.? Is it correct?

Reply: Thanks for your suggestion. We have corrected this error.

10. Line 414. Where is the Table 5B???

Reply: Thanks for your suggestion. We have carefully checked and corrected the error, and we changed Table 5 to Supplement table 6.

Discussion:

Discussion section is too lengthy and need to be largely revised. The authors should delete many descriptive expressions. Most importantly, authors need to strengthen the logicity of the discussion. Additionally, the discussion section does not require so many sub-titles, short and concise titles should be summarized in this section.

Rely: Thanks for your suggestion. we have deleted many descriptive expressions. We have rewritten the Discussion made it shortly and concise.

1. Line 427. microbes

Reply: we have corrected the “microbe” to “microbes”

2. Line428. various species? There are only two different types here.

Reply: we have changed the “various” to “two”

3. Line 448. Reword this sentence.

Reply: Thanks for your suggestion. We have reworded this sentence.

4. Lines 449-452. Why did the authors get this conclusion? Add some related results and references to enhance your statements.

Reply: Thanks for your suggestion. We have cited inappropriate references, and we have reorganized the literature to rectify this paragraph.

5. Lines 452-454. The authors need to elucidate why and how the previous studies' findings consistent with your results.

Reply: Thanks for your suggestion. We have cited inappropriate references, and we have reorganized the literature to rectify this paragraph.

6. Line 465. What is the purpose of introducing “core microbiome” here? Dose it relevant to your results?

Reply: Thanks for your suggestion. We explored the species with high network centrality and importance in each sample as the key species or core microorganisms, and infer the possible ecological role of the key species on the salt tolerance of *Suaeda*. We have changed this content.

7. Line 474. Why did the authors discuss the pathogen suppressive of Actinobacteria? I did not see any results about the pathogens in your results. Thus, I suggest to carefully reconsider the logicity of these sentences.

Reply: Thanks for your suggestion. We have delated the inappropriate references.

8. Line 477-478. While Actinobacteria and Proteobacteria are well-known dominant bacterial taxa in soils and are known to contribute to plant growth and health, this study focuses on specific bacterial groups identified as keystone taxa in saline soils. These keystone taxa have the potential to play critical functional roles, and we are particularly interested in exploring their potential functions in promoting plant growth and health in these challenging environments.

Reply: Thanks for your suggestion. We have added this sentence in this paragraph.

9. Line 501-502. should include the relevant results and discuss more about this taxa to explain the importance.

Reply: Thanks for your suggestion. We have added more discuss and reference about this taxa to explain the importance

10. Line 508. “Marmoricola” should be italic.

Reply: Thanks for your suggestion. We have corrected the “Marmoricola” in italic.

11. Line 519. They are often found in...

Reply: Thanks for your suggestion. We have corrected this sentence.

12. Line 522-526. Reword this sentence.

Reply: Thanks for your suggestion. We have reworded this sentence.

13. Line 529-530. Cited the relevant reference(s).

Reply: Thanks for your suggestion. We have added a relevant reference.

14. Line 552-554. It's hard to understand the purpose of this paragraph and its connection with above.

Reply: Thanks for your suggestion. We have reworded this paragraph.

15. Line 557-568. Soil pH is an important factor that driving soil microbial community assembly. However, in this study, authors should focus on the soil salinization.

Reply: Thanks for your suggestion. We have reworded this paragraph, and focus on the soil salinization.

16. Line 574. What is "t microbial network"?

Reply: We have corrected the "t microbial network" to "microbial network".

17. Line 579-580. What is the relationship between these physical constraints and microbial networks? These sentences are vague and need to be reword.

Reply: Thanks for your suggestion. We have reworded this sentence.

18. Lines 584-590. The authors' hypotheses require further elaboration and deeper discussion. It is unclear how they arrived at these hypotheses, and more statements and evidence are needed to support their claims.

Reply: Thanks for your suggestion. We have added more statements and evidence to elaborate our hypotheses in "Unearthed of keystone taxa in the rhizosphere of *Suaeda*" section.

19. Line 595. soil's network is vague.

Reply: we have corrected the "soil's network" to "soil's microbial network".

August 30, 2023

Prof. Yan fei Sun
Shihezi University
Shihezi
China

Re: Spectrum01649-23R1 (Insight into bacterial and archaeal community structure of *Suaeda altissima* and *Suaeda dendroides* rhizosphere in response to different salinity level)

Dear Prof. Yan fei Sun:

Thank you for submitting your manuscript to Microbiology Spectrum. As you will see your paper is very close to acceptance. Please modify the manuscript along the lines I have recommended. As these revisions are quite minor, I expect that you should be able to turn in the revised paper in less than 30 days, if not sooner. If your manuscript was reviewed, you will find the reviewers' comments below.

When submitting the revised version of your paper, please provide (1) point-by-point responses to the issues raised by the reviewers as file type "Response to Reviewers," not in your cover letter, and (2) a PDF file that indicates the changes from the original submission (by highlighting or underlining the changes) as file type "Marked Up Manuscript - For Review Only". Please use this link to submit your revised manuscript. Detailed instructions on submitting your revised paper are below.

Link Not Available

Sincerely,

Jing Han

Reviewer comments:

Reviewer #1 (Comments for the Author):

The authors have addressed my comments well and the manuscript has greatly improved. However, I still have some concerns about the paper as detailed below. Also, the authors should indicate the revised line number for easier viewing by the reviewers.

1. The novelty of abstract need to be improved.
2. L27-28 What's the authors mean that pH was the important factors of soil characteristics?
3. L88-91. Enhanced the logicity of these sentences.
4. L342, L353, L407, L422, L445, L528, L540. Font format. Please carefully check the whole manuscript.
5. L369. Influenced or impacted.
6. L577. Reword.
7. L531. What's this mean?

Preparing Revision Guidelines

Please return the manuscript within 60 days; if you cannot complete the modification within this time period, please contact me. If you do not wish to modify the manuscript and prefer to submit it to another journal, please notify me of your decision immediately so that the manuscript may be formally withdrawn from consideration by Microbiology Spectrum.

The authors have addressed my comments well and the manuscript has greatly improved. However, I still have some concerns about the paper as detailed below. Also, the authors should indicate the revised line number for easier viewing by the reviewers.

1. The novelty of abstract need to be improved.
2. L27-28 What's the authors mean that pH was the important factors of soil characteristics?
3. L88-91. Enhanced the logicality of these sentences.
4. L342, L353, L407, L422, L445, L528, L540. Font format. Please carefully check the whole manuscript.
5. L369. Influenced or impacted.
6. L577. Reword.
7. L531. What's this mean?

Dear reviewer:

Thank you for giving us the opportunity to submit a revised draft of the manuscript “Analysis of bacterial and archaeal community structure in *Suaeda altissima* and *Suaeda dendroides* rhizosphere under different salinity level” (Spectrum01649-23R1) for publication in *Microbiology Spectrum*. We appreciate the time and effort that you dedicated to providing feedback on our manuscript and are grateful for the insightful comments on and valuable improvements to our paper. We have incorporated the suggestions made by the reviewers. Please see below for a point-by-point response to the reviewers’ comments and concerns.

1. The novelty of abstract need to be improved.

Reply: Thanks for your suggestion. We have modified this abstract, and marked yellow.

2. L27-28 What’s the authors mean that pH was the important factors of soil characteristics?

Reply: Thanks for your reminder. What we mean by this statement is: pH was one of the most important drive factors of soil characteristics to shaped bacterial and archaeal community structure. we have modified this sentence, and marked yellow. (line 17)

3. L88-91. Enhanced the logicity of these sentences.

Reply: Thanks for your suggestion. We have modified this sentence, and enhanced the logicity of these sentences. (line 62-65)

4. L342, L353, L407, L422, L445, L528, L540. Font format. Please carefully check the whole manuscript.

Reply: Thanks for your reminder. We have checked the Font format in whole manuscript.

5. L369. Influenced or impacted.

Reply: Thanks for your suggestion. We have changed the “Predominant impacts” to “Influenced”. (line 280)

6. L577. Reword.

Reply: Thanks for your suggestion. We have reworded this sentence. (line 446-450)

7. L531. What’s this mean?

Reply: Thanks for your reminder. Due to our mistake, we wrote the wrong sentence. We have removed this sentence. (line 410)

September 8, 2023

Prof. Yan fei Sun
Shihezi University
Shihezi
China

Re: Spectrum01649-23R2 (Insight into bacterial and archaeal community structure of *Suaeda altissima* and *Suaeda dendroides* rhizosphere in response to different salinity level)

Dear Prof. Yan fei Sun:

The authors should make the following minor changes.

1. Line 281, "Influenced of soil pH" should be "Effect of soil pH".
2. Line 448-452, no visible change is found between the reworded new sentences and the deleted one. Line 451, delete "The" from "The more lines".

Your manuscript has been accepted, and I am forwarding it to the ASM Journals Department for publication. You will be notified when your proofs are ready to be viewed.

Sincerely,

Jing Han
Editor, Microbiology Spectrum
